# TopicSummRAG: A Topic-Enhanced RAG model for Improving Query-Focused Summarization from Long Documents

## Abstract

Despite larger context windows, large language models (LLMs) continue to struggle with answering queries over long, unstructured documents. Retrieval-Augmented Generation (RAG) mitigates this limitation by retrieving relevant text before generation; however, its effectiveness depends critically on how documents are segmented into retrievable units. Existing chunking strategies—such as fixed-size or sliding windows—often ignore topical coherence, frequently merging unrelated content or fragmenting coherent discourse, which degrades retrieval precision and downstream generation quality. We propose TopicSummRAG, a framework that aligns retrieval units with the latent topical structure of documents. TopicSummRAG first segments documents into contextually coherent topical chunks using a boundary-supervision-free contrastive text segmentation model, and then summarises each segment to form compact retrieval metadata. The segmentation component is evaluated independently on the QMSum and TIAGE benchmarks, where it consistently improves boundary detection and placement over strong baselines. During retrieval, segment-level summaries are matched against the query, and an entropy- and dominance-based filtering strategy adaptively selects relevant segments by measuring the concentration of relevance mass, avoiding brittle fixed cutoffs. We evaluate TopicSummRAG on ODSum-Story, ODSum-Meeting, and QMSum across multiple retrievers (BM25, SBERT, Situated) and LLMs (Qwen3-8B, LLaMA-3.1-8B, Gemma-3-12B). TopicSummRAG yields improvements of up to 13% ROUGE-1, 25% ROUGE-2, and 20% ROUGE-L, alongside 10–15 point gains in LENS and up to 10-point improvements in BLANC. These results demonstrate that topic-aware segmentation and adaptive retrieval substantially improve factual grounding, coherence, and retrieval robustness, providing a retriever- and model-agnostic framework for long-document query-focused summarisation and question answering with RAG.

## 1 Introduction

Despite advances in large language models (LLMs) and their ability to process extended contexts, query-focused information access over long, multi-topic documents remains a challenging task. Tasks such as query-focused summarisation (QFS) or question answering often involve long reports or conversational transcripts that span multiple topics and discourse phases. While LLMs can ingest long contexts, they lack an inherent mechanism to isolate query-relevant regions, frequently resulting in diffuse attention, reduced factual precision, or hallucinated content. Retrieval-Augmented Generation (RAG) mitigates this issue by retrieving relevant text before generation, but its effectiveness critically depends on document segmentation. Most RAG systems rely on fixed-size chunks, sliding windows, or sentence-level units that ignore latent topical structure. Such heuristics merge unrelated content or fragment coherent discourse, degrading retrieval quality—an issue amplified in long documents with abrupt shifts and gradual topical drift.

The notion of a topic varies substantially across NLP subfields. Latent topic models define topics as distributions of words, while dense retrieval encodes topics implicitly in embedding spaces. We instead define topics as contiguous discourse units, spans of text that are internally coherent and separated by semantic shifts. In this work, we propose TopicSummRAG, a framework that aligns retrieval units with the latent topical organisation of long documents. TopicSummRAG segments documents into contextually coherent topical

Table 1: Comparison of generic RAG and TopicSummRAG on a QMSum query

| Query | Why did the team choose single-curved design when discussing remote control style? |
|---|---|
| **Gold Span IDs** | $[185\text{–}188], [372\text{–}381]$ |
| **RAG Span IDs** | $[163, 153, 287, 187]$ |
| **TopicSummRAG Span IDs** | $[185, 187, 375, 376, 379, 381]$ |
| **Gold Summary** | Industrial Designer introduced uncurved, single-curved and double-curved designs. The team found the uncurved design too dull, and the double-curved design limited functionality (e.g., scroll-wheels could not be used). Therefore, the team selected the single-curved design as a balance. |
| **Sentence-level RAG** | The team chose a single-curved design for the remote control because it allows for a comfortable grip and placement of buttons within reach of the user's thumb. |
| **TopicSummRAG** | The team evaluated flat, double-curved, and more complex designs. A flat design was seen as too dull, while double-curved limited interaction options (e.g., prevented use of scroll-wheels). Hence, they chose a single-curved design as a balance between visual appeal and usability. |

spans using a learned segmentation model and summarises each segment to form compact retrieval metadata. During inference, queries are matched against these segment-level summaries to select discourse-consistent regions for generation.

Table 1 shows an example that highlights the limitations of generic RAG and the benefit of TopicSummRAG on a query from QMSum meeting transcript. In this example, the query is about "Why did the team choose single-curved design when discussing remote control style?". In sentence-level RAG, the retriever selects individual sentences based on their semantic similarity to the query. From a query-focused summarization perspective, this leads to retrieving sentences that are relevant in isolation but do not collectively support a complete or accurate answer. In this example, RAG retrieves spans that fail to capture the essential evidence needed for the query. This highlights a common limitation where RAG retrieves broadly relevant content but often misses the specific evidence spans most aligned with the query intent, resulting in incomplete query-focused summaries. In contrast, TopicSummRAG segments the document into coherent units and retrieves them using segment-level summaries. This makes it easier to aggregate query-relevant evidence spans across the document, such as why the uncurved design was rejected (too dull) and why the double-curved design was not feasible (it prevents scroll-wheels across different discourse segments. As each segment preserves its context, the retrieved evidence achieves better query coverage and reduced fragmentation. Hence, query-focused summary generated by TopicSummRAG better reflects the underlying decision rationale.

We evaluate our segmentation model on TIAGE (Xie et al., 2021) and QMSum (Zhong et al., 2021), and evaluate TopicSummRAG on multiple QFS benchmarks, including ODSum-Story, ODSum-Meeting (Rao et al., 2024), and QMSum, against strong RAG baselines. Across datasets, TopicSummRAG consistently improves retrieval accuracy and factual faithfulness while substantially reducing hallucinations. Overall, we contribute (i) a novel topic segmentation model for long documents and (ii) TopicSummRAG, a retrieval-augmented summarisation framework that improves RAG performance through topic-aware chunking and segment-level summaries, significantly outperforming state-of-the-art methods on query-focused long-document summarisation benchmarks.

## 2 Related Work

**Query-Focused Summarization** QFS aims to generate summaries that directly satisfy an explicit information need rather than providing a generic overview. Early extractive and abstractive approaches relied on lexical salience and query–document similarity (Baumel et al., 2016; 2018), which perform well on short or topically concentrated texts but degrade on long, multi-topic documents. Domain-specific QFS systems in areas such as biomedicine and law emphasise grounded evidence and factual consistency (Sankhavara &

Majumder, 2020; Conrad et al., 2009). Recent query-based RAG methods (e.g., QB-RAG) improve alignment by matching queries against structured query representations (Yang et al., 2025a). However, most approaches remain constrained by their retrieval units, typically operating at sentence or turn granularity or over truncated contexts, which disconnects retrieved evidence from the broader discourse structure required for coherent summarisation.

**Retrieval-Augmented Generation and Retrieval Units** Retrieval-augmented generation (RAG) grounds generation in external evidence to reduce hallucinations (Lewis et al., 2020; Brown et al., 2025). Extensions incorporating structured or graph-based retrieval improve global reasoning but still rely on heuristic segmentation and weak topic alignment (Edge et al., 2025; Hong et al., 2025). Consequently, RAG-based QFS systems face three recurring issues: (i) a precision–coverage trade-off as retrieval units grow; (ii) semantic misalignment between lexically similar but topically distinct spans; and (iii) scalability constraints when costly segmentation is applied at inference. Recent work shows that segmentation quality critically affects retrieval and downstream performance (Wang et al., 2025). TopicSummRAG addresses these limitations by using discourse-coherent segments and segment-level summaries as retrieval metadata. While segmentation is query-agnostic, retrieval is explicitly query-conditioned through summary-based matching and selection.

**Topic Segmentation and Chunking** Fixed-size and sentence-level chunking are efficient but often disrupt discourse coherence (Yepes et al., 2024). Classical unsupervised methods (e.g., TextTiling, C99) rely on lexical overlap and struggle with contextual coherence (Hearst, 1997). Recent approaches leverage contextual embeddings and neural models to better capture discourse structure (Song et al., 2022; Koshorek et al., 2018; Nair et al., 2023). Very recent work optimises chunking for RAG using learned or LLM-informed partitioning (Duarte et al., 2024; Zhao et al., 2025; Günther et al., 2025), but typically treats segmentation and retrieval as loosely coupled.

TopicSummRAG jointly addresses segmentation, retrieval, and summarisation by learning discourse-aware topical segments, summarising them into compact retrieval units, and applying adaptive, dominance-based selection. This design explicitly controls retrieval granularity, enabling coherent and recall-oriented query-focused retrieval over long documents.

## 3    Methodology

We study the problem of query-focused retrieval and summarisation over long, multi-topic documents. Given a document $D = \{u_1, u_2, \ldots, u_n\}$ composed of $n$ textual units, such as sentences, paragraphs, or dialogue turns and a user query $q$, the goal is to retrieve a compact set of evidence chunks $C = \{c_1, c_2, \ldots, c_m\}$ that collectively satisfy the information need expressed in $q$, and to generate a grounded, coherent response. A key challenge in this setting is that long documents often exhibit both abrupt topic transitions and gradual topical drift, while lacking explicit boundary markers. As a result, fixed-size or heuristic chunking strategies often fail to align retrieval units with the underlying semantic structure, leading to imprecise retrieval and degraded generation quality.

To address these challenges, we propose TopicSummRAG (Figure 5), a two-stage framework that explicitly aligns retrieval units with the latent organisation of long documents. The framework first segments a document into coherent spans and then performs topic-guided retrieval and summarisation over these spans. The first stage identifies contiguous sequences of units that form cohesive discourse segments (Section 3.1). These segments represent the smallest units that are internally consistent, and may span sentences, paragraphs, or dialogue turns. Let such a unit be denoted as $s_i$. In the second stage, these units are treated as retrieval chunks for TopicSummRAG (Section 3.2). Overall, TopicSummRAG performs retrieval and generation in five stages: (1) segmentation, (2) coarse retrieval using segment-level summaries, (3) entropy-based adaptive selection, (4) fine-grained sentence-level retrieval within selected regions, and (5) final answer generation. A detailed pipeline with examples is provided in Appendix A.3.

Importantly, our goal is not to propose a fundamentally new segmentation algorithm, but to adapt segmentation for query-focused summarization, where segments are evaluated based on their usefulness for retrieval and generation. As we show in Appendix A.11 and A.12, this formulation has a direct impact on both retrieval quality and downstream summaries.

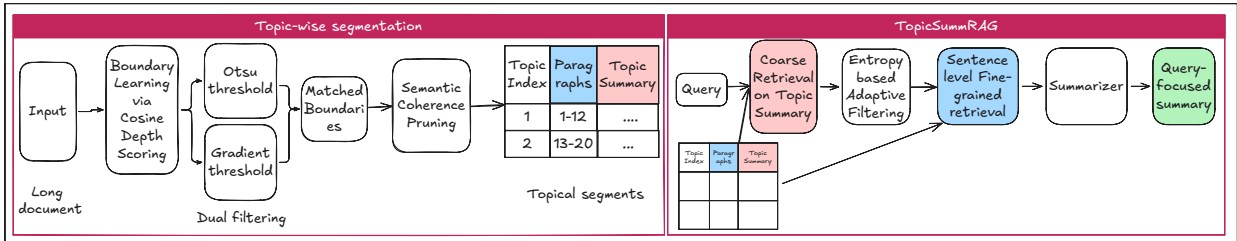

Figure 1: Pipeline of TopicSummRAG

### 3.1 Topical segmentation

We conceptualise "topics" as contiguous discourse segments, in simpler terms, locally cohesive spans of text that are internally consistent and meaningfully different from their neighbors. In this sense, our notion of a topic is closer to discourse segmentation than to classical topic modeling, as it does not assume globally defined or labeled topics. When retrieval units correspond to such segments, relevance scoring becomes more stable and retrieved content is more likely to support grounded generation. This motivates our decision to treat segmentation as a first-class component, rather than as a downstream heuristic.

A key challenge in segmentation is that boundary detection is inherently contextual. Topic transitions are better characterised by changes in contextual semantics than by surface-level cues. While abrupt transitions (e.g., agenda changes in meetings or plot shifts in narratives) often produce sharp semantic breaks, many documents exhibit gradual drift, where relevance evolves slowly across units. As a result, whether a unit signals continuation or shift cannot be determined in isolation, but only relative to its surrounding context. Two adjacent units may appear similar yet belong to different segments, while units within the same segment may vary lexically. This makes simple similarity measures insufficient for reliable boundary detection. Although documents are composed of structural units such as sentences or dialogue turns, their semantic organization is governed by higher-level structures such as subtopics or discussion threads, which are not explicitly marked.

Our approach addresses this by identifying boundaries in a data-driven and domain-general manner. While the underlying signal builds on similarity-based methods, we combine contextual embeddings with global and local filtering strategies to improve robustness in long documents. More importantly, segmentation is integrated into a retrieval pipeline where its effectiveness is evaluated through downstream performance rather than standalone boundary accuracy.

**Learning Contextual Embeddings:** Standard text encoders produce representations based on local content similarity, but they are not trained to distinguish semantic continuity from semantic change across discourse. To create embeddings that can account for topical coherence, we introduce a context-aware contrastive pre-training framework. Using *unsup-simcse-roberta-base*[1], each unit $s_i$ (turn, sentence, or paragraph depending on domain) is encoded as $h_i = f_\theta(s_i)$, where $\theta$ is the parameter of the network learned via a triplet loss function. For each anchor unit $s_i$, the positive example $s^+$ is drawn from the same topical span (i.e., semantically aligned neighbouring units) and the negative example $s^-$ is sampled from a different span, representing a topic shift. The triplet loss is defined as: $\mathcal{L}_{\text{triplet}} = \sum_{(s,s^+,s^-)} \max\left(0,\ m + d(s, s^+) - d(s, s^-)\right)$, where $d$ is the cosine distance and $m = 0.5$. Triplets are constructed using hard negative mining: the most similar in-span unit serves as the positive. In contrast, semantically close out-of-span units are selected as negatives (Figure 4 in Appendix A.2). This encourages the encoder to separate semantically similar but topically distinct units, facilitating precise boundary detection. Encoding such contextual distinctions is critical not only for accurate boundary detection but also for ensuring that downstream retrieval units align with coherent discourse regions rather than isolated sentences. Appendix A.1 shows the segmentation pipeline .

---

[1] princeton-nlp/unsup-simcse-roberta-base

**Boundary Learning using Cosine Depth Scoring:** The contextual embeddings learned in the previous step provide a representation space in which semantic continuity and semantic change are explicitly separated. Leveraging these representations, we detect topical boundaries by measuring the sharpness of contextual semantics change across adjacent regions of the document, rather than between individual units.

Specifically, we identify candidate topic boundaries using cosine-depth scoring, which measures semantic discontinuity between left and right contextual windows. For each unit $i$ (sentence, paragraph, or dialogue turn), we compute left and right contextual embeddings $e_i^{\text{left}}$ and $e_i^{\text{right}}$ by pooling augmented unit embeddings within a fixed window of size $w$. The semantic depth score is defined as $\text{depth}_i = 1 - \frac{\mathbf{e}_i^{\text{left}} \cdot \mathbf{e}_i^{\text{right}}}{\|\mathbf{e}_i^{\text{left}}\| \, \|\mathbf{e}_i^{\text{right}}\|}$. By aggregating context on both sides of a position, cosine depth captures both abrupt semantic breaks and gradual topical drift. However, depth signals alone are insufficient, as long documents naturally contain noise and local fluctuations that can produce spurious peaks.

Empirically, we observe that meaningful topic boundaries exhibit two key properties: (i) *global salience*, where the depth score stands out relative to the overall distribution, and (ii) *local abruptness*, corresponding to sharp semantic transitions rather than gradual drift. To capture global salience, we apply Otsu thresholding (Otsu, 1979) on the cosine-depth values. Otsu's method automatically determines a threshold that separates the depth distribution into two groups by maximizing inter-class variance. In our setting, this corresponds to partitioning positions into *low-depth* (within-topic continuity) and *high-depth* (potential topic boundaries). This eliminates the need for manual threshold tuning and adapts dynamically to variations across documents. Given the set of depth scores $\{\text{depth}_i\}$, we compute a global threshold $\tau_{\text{Otsu}}$, and retain only those positions where $\text{depth}_i \geq \tau_{\text{Otsu}}$, ensuring that selected candidates are globally prominent. However, global prominence alone is insufficient, as gradual topic drift may also yield high depth values. To enforce local abruptness, we compute the first-order difference of the depth curve, $\Delta_i = |\text{depth}_i - \text{depth}_{i-1}|$, and retain positions with high gradient magnitude, corresponding to sharp semantic changes. Finally, since these signals may not perfectly align due to contextual windowing, we reconcile candidates within a small tolerance window ($\pm 1$–2 positions). A position is selected as a topic boundary only if it satisfies both criteria, being globally salient under Otsu thresholding and locally abrupt under gradient filtering—resulting in a robust set of topic boundaries.

**Semantic Coherence Pruning:** Even after dual thresholding, candidate boundaries may remain overly dense or redundant, particularly in documents with smooth topical flow. Enforcing a fixed segmentation granularity is undesirable, as different documents—and even different regions within the same document—exhibit varying structural complexity. We therefore introduce semantic coherence pruning to determine whether introducing a candidate boundary would meaningfully improve segment homogeneity. Rather than relying on external supervision or fixed length constraints, coherence pruning uses embedding-based consistency to decide whether a segment should be split.

For a candidate boundary that splits units $i{:}j$, coherence is defined as $\text{coh}(i,j) = \frac{1}{j-i+1} \sum_{k=i}^{j} \frac{\|h_k'\|_2}{\frac{1}{j-i+1} \sum_{l=i}^{j} \|h_l'\|_2 + \epsilon}$, where $h_k'$ denotes the feature-augmented embedding of unit $k$, $\|\cdot\|_2$ is the $\ell_2$ norm, and $\epsilon$ is a small constant for numerical stability. The numerator measures the semantic strength of each unit relative to the segment, while the denominator normalizes by the average embedding magnitude across the segment. This measure captures the internal semantic homogeneity of a segment: high coherence indicates a semantically consistent region that should be preserved, whereas low coherence suggests topical drift and motivates splitting. Here, the $\ell_2$ norm of the feature-augmented embedding $|h_k'|_2$ serves as a proxy for the semantic strength of a unit. Since the encoder is trained with a contrastive objective to distinguish intra-topic and inter-topic units, embedding magnitudes tend to reflect alignment with the surrounding discourse. As a result, semantically coherent segments exhibit stable embedding norms, while segments containing topic drift show higher variance. The proposed normalization therefore captures intra-segment consistency in an unsupervised manner.

We apply coherence pruning sequentially after cosine-depth-based candidate selection to regulate final segmentation granularity. To adapt pruning strength across documents, we compute a document-level homogeneity score defined as the ratio of the depth curve's standard deviation to its mean. Documents with smooth topical flow undergo more aggressive pruning, yielding fewer and broader segments, while documents

with frequent sharp transitions are pruned conservatively to preserve finer-grained boundaries. Together, cosine-depth scoring determines where boundaries may occur, while coherence pruning decides which of these candidates are retained, producing segmentations that are precise, semantically coherent, and adaptable across document types.

### 3.2 Topic guided retrieval and summarization

Having segmented the document into semantically coherent topical spans, the remaining challenge is to efficiently identify which of these spans are relevant to a given query. Direct retrieval over full segments can be sensitive to surface-level variation and inefficient when documents contain many topics. Instead, we perform coarse retrieval over compact segment-level summaries, which serve as high-level semantic abstractions of each topical region. This design allows retrieval to operate at the level of discourse topics rather than individual sentences, improving robustness to lexical noise while preserving thematic relevance. Moreover, operating over summaries yields a compact relevance distribution that can be analyzed to adapt retrieval breadth based on query ambiguity.

TopicSummRAG is retriever-agnostic: it operates with any model capable of producing vector representations for segment summaries or sentences. The framework requires only a similarity function (e.g., cosine similarity) for ranking units during both coarse- and fine-grained retrieval. For clarity, we describe the method assuming a dense encoder such as SBERT, though the retrieval logic remains unchanged for sparse retrievers (e.g., BM25) or discourse-aware models (e.g., Situated Embeddings). The full set of retrievers and generators evaluated is described in Section 4.2.

**Topic-aware Coarse Retrieval using Segment-level Summaries:** Using our segmentation method (Section 3.1), the document is divided into topical segments, each representing a semantically coherent discourse region. Each segment is summarized using Qwen3-8B.[2] These summaries provide compact, topic-level representations that abstract away sentence-level variability while preserving dominant semantic content. We embed the query and all segment summaries into a shared semantic space using SBERT and compute cosine similarities. The top-$k$ summaries are retained as high-level topical filters, narrowing the search space to segments thematically aligned with the query.

Following this coarse ranking, we retain the top-$L$ segment summaries as a recall-oriented candidate set. The parameter $L$ defines a fixed upper bound on the number of segments considered for further processing and is chosen to ensure coverage rather than precision. Importantly, $L$ does not determine the final retrieval granularity; it merely bounds the search space prior to adaptive selection. The effective number of retrieved segments is determined in the subsequent stage based on the structure of the relevance distribution. Because adaptive selection depends on relative dominance rather than absolute rank, the same thresholds remain valid as $L$ varies.

**Entropy-Based Adaptive Selection:** Following topic-aware coarse retrieval, we obtain a ranked list of segment-level summaries with cosine similarity scores relative to the query. However, the sharpness of this similarity distribution varies substantially across queries and documents: some queries are tightly aligned with a single topical segment, while others are inherently broad and span multiple segments. Applying a fixed retrieval cutoff in such cases is brittle and can either discard relevant context or introduce excessive noise. To address this, we analyze the distribution of segment relevance scores using a temperature-scaled softmax. Because cosine similarities lie in a compressed numeric range, directly applying softmax can yield overly flat probability distributions even when meaningful relevance gaps exist. We therefore apply temperature scaling with $\tau = 0.3$ prior to normalization, restoring contrast in the similarity distribution in line with standard practice in contrastive retrieval models. Given top-$L$ similarity scores $\{s_1, \ldots, s_L\}$, we compute: $p_i = \frac{\exp(s_i/\tau)}{\sum_{j=1}^{L} \exp(s_j/\tau)}$. We compute the entropy of this distribution, $H = -\sum_{i=1}^{L} p_i \log p_i$, and its normalized form $H_{\mathrm{norm}} = H/\log L$, which serves as a measure of uncertainty in segment relevance. While entropy captures the overall dispersion of relevance mass, we do not use it directly to determine the number of retrieved segments. Instead, the decisive signal is the dominance of the most relevant segment, quantified as $p_{\max} = \max_i p_i$, which measures how much of the total relevance mass is explained by a single topical region.

---

[2]https://huggingface.co/Qwen/Qwen3-8B

Based on the dominance of the most relevant segment, measured by the maximum softmax probability $p_{max}$, we adaptively select the number of retrieved segments k. For focused queries ($p_{max} \geq 0.6$), a single segment accounts for the majority of the relevance mass, and we retain only the top segment (k=1). For weakly multi-topic queries ($0.4 \leq p_{max} < 0.6$), one dominant segment and a secondary competitor are present, and we retain the top two segments (k=2). For broad queries ($p_{max} < 0.4$), relevance is distributed across multiple segments, and we retain all top-L segments to preserve coverage.

Under $\tau = 0.3$, these thresholds correspond to meaningful odds ratios between the top segment and its competitors, distinguishing single-topic, near-single-topic, and genuinely multi-topic queries in a distributional rather than heuristic manner. Entropy thus provides a complementary diagnostic signal reflecting uncertainty, while dominance governs the strength of retrieval pruning. This adaptive selection strategy enables TopicSummRAG to remain precise for focused queries while preserving coverage for broader information needs.

**Sentence-level fine-grained retrieval within matched regions:** Once relevant segments are identified, we perform second-stage sentence-level retrieval to extract the most evidence-rich text. Each selected segment is decomposed into sentences, forming a fine-grained retrieval corpus $S = \{s_1, s_2, \ldots, s_L\}$. By default, we use SBERT as our dense retriever (we also evaluate BM25 and Situated Embeddings in Section 4.2). The query embedding $q$ is compared against all sentence embeddings using cosine similarity. We rank sentences and select the top-$k_s$ evidence sentences (we use $k_s = 10$). This produces a targeted evidence set: $E = \{e_1, e_2, \ldots, e_{k_s}\}$ where each $e_i$ is a highly relevant sentence grounded in both local topic and global query semantics.

Topic-level summaries filter which regions matter. Sentence-level retrieval identifies what exact information is relevant inside those regions. This two-tier design balances high recall (via segment selection) with high precision (via sentence retrieval), which was not possible with chunk-level RAG alone.

**Final answer generation:** We concatenate the retrieved evidence sentences with the query and provide them to a summarizer model for final answer generation (prompt added in Appendix A.7). The model is instructed to generate an answer grounded strictly in retrieved evidence, mitigating hallucination and over-retrieval. The final evidence set thus contains the best matching topical regions and the most relevant sentences within those regions. Together, these ensure that the generator conditions its output on coherent, query-focused, and contextually grounded content.

## 4 Experiments and Results

We present results on two sets of experiments - Experiment 1 shows our results on the topic segmentation model and Experiment 2 on query-focused summarization.

### 4.1 Experiment 1 - Topic segmentation

#### 4.1.1 Datasets:

We evaluate our topic segmentation model on two distinct benchmarks that capture different aspects of conversational dynamics: open-domain social chitchat (TIAGE (Xie et al., 2021)) and multi-party formal meetings (QMSum (Zhong et al., 2021)). The detailed statistics for both datasets are summarised in Table 2 and further details are added in Appendix A.4.1.

#### 4.1.2 Baselines:

To evaluate the effectiveness of our proposed unsupervised segmentation approach, we compare against a comprehensive set of baseline models listed below.

- Fine-tuned / Supervised

Table 2: Statistics of (a) topic segmentation datasets (Left) and (b) query-focused summarization datasets (Right). *Tr = Train, Va = Validation, Te = Test; W = Words, U = Utterances, Doc = Document, Q = Query, L = Length.*

| Dataset | # Docs | # Qs | # Pairs |
|---|---|---|---|
| ODSum-story | 1190 | 635 | – |
| ODSum-meeting | 232 | 436 | – |
| QMSum (All) | 232 | 7.8 | 279 |
| Dataset | Avg. Doc L. | Avg. Q. L. | Avg. Sum. L. |
| ODSum-story | 808.54 | 10.79 | 273.80 |
| ODSum-meeting | 7176.21 | 32.89 | 185.17 |
| QMSum (All) | 9069.8 | – | 69.6 |

| Dataset | # Docs | | | # Words/Doc | | | Avg. | |
|---|---|---|---|---|---|---|---|---|
| | Tr | Va | Te | Min | Avg | Max | W/Section | U/Doc |
| TIAGE | 286 | 96 | 97 | 109 | 185.1 | 264 | 40.4 | 15.4 |
| QMSum | 162 | 35 | 35 | 1371 | 9521.4 | 25529 | 1593.6 | 334.7 |

- BiLSTM_RoBERTa (Ghinassi et al., 2023): A hybrid encoder that combines RoBERTa embeddings with a bidirectional LSTM layer to capture both contextual token representations and sequential discourse dependencies.
- TopSeg_RoBERTa (Ghinassi et al., 2023): TopSeg applies the DeepTiling (Ghinassi, 2021) algorithm on sentence embeddings generated by RoBERTa.
- MiniSeg (Retkowski & Waibel, 2024): A hierarchical text segmentation model originally designed for smart chaptering of unstructured spoken content.
- Deep Tiling with Sentence Embeddings: In addition to reported baselines, we apply the deep tiling segmentation technique using two widely adopted sentence-transformer models `all-MiniLM-L6-v2`[3] and `all-mpnet-base-v2`[4].
- DialogLM (Zhong et al., 2022): A dialogue-specific pretrained language model designed for meeting summarization and segmentation tasks.
- UPS (Yang et al., 2025b): A supervised segmentation model explicitly pre-trained on the TIAGE distribution via Utterance-Pair Modeling. UPS transforms topic segmentation into a binary classification of utterance pairs rather than global sequence labeling.

- Untuned / Unsupervised

  - SumSeg (Artemiev et al., 2024): An unsupervised dialogue topic segmentation method that leverages summarization signals for boundary detection.
  - Gemini: We used `gemini-1.5-flash` in a zero-shot setting, where segmentation is obtained by directly prompting the model.

For SumSeg, MiniSeg, DialogLM and UPS, we report segmentation results directly from the respective papers to ensure faithful comparison with published benchmarks. For other baselines, we have conducted experiments locally under consistent settings using publicly available code and models.

### 4.1.3 Metrics:

To assess the performance of our segmentation model, we employ a combination of boundary-based classification metrics (Precision, Recall, and the F1-score) and distance-based structural metrics ($P_k$ metric (Beeferman et al., 1999) and WindowDiff (Pevzner & Hearst, 2002)). This dual evaluation captures both boundary-level accuracy and global structural consistency, which is essential for assessing segmentation quality in long, multi-topic documents. Further details are added in Appendix A.4.2.

### 4.1.4 Results

Table 3 reports segmentation results on both the QMSum and TIAGE datasets. On QMSum, our approach achieves the highest F1 score (0.263) and the highest recall (0.5631), indicating that it recovers a larger proportion of true segment boundaries while maintaining substantially lower boundary error rates.. Although

---

[3]https://huggingface.co/sentence-transformers/all-MiniLM-L6-v2
[4]https://huggingface.co/sentence-transformers/all-mpnet-base-v2

Table 3: Results of topic segmentation models on QMSum (Left) and TIAGE (Right) datasets. * indicates results reported directly from the respective papers. *Note:* F1 is computed as the macro-average of per-document F1 scores

| Model | P | R | F1 | $P_k$ | WD |
|---|---|---|---|---|---|
| *Fine-tuned / Supervised* | | | | | |
| BiLSTM | 0.0644 | 0.1406 | 0.0313 | 0.4121 | 0.4830 |
| TopSeg | 0.0134 | 0.0305 | 0.0181 | 0.3531 | 0.3830 |
| MiniLM | 0.0290 | 0.0240 | 0.0260 | 0.4580 | 0.4830 |
| Mpnet | 0.0310 | 0.0280 | 0.0300 | 0.4370 | 0.4660 |
| MiniSeg* | **0.3109** | 0.1692 | 0.2192 | – | – |
| DialogLM* | – | – | – | 0.380 | 0.400 |
| *Untuned / Unsupervised* | | | | | |
| SumSeg* | – | – | – | 0.357 | 0.379 |
| Gemini | 0.0740 | 0.3600 | 0.1170 | 0.2640 | 0.3070 |
| Ours | 0.1716 | **0.5631** | **0.2630** | **0.3579** | **0.3789** |

| Model | P | R | F1 | $P_k$ | WD |
|---|---|---|---|---|---|
| *Fine-tuned / Supervised* | | | | | |
| BiLSTM | 0.0580 | 0.1280 | 0.0790 | 0.440 | 0.500 |
| TopSeg | 0.0120 | 0.0270 | 0.0170 | 0.390 | 0.410 |
| MiniLM | 0.0260 | 0.0220 | 0.0240 | 0.480 | 0.505 |
| Mpnet | 0.0280 | 0.0250 | 0.0260 | 0.460 | 0.490 |
| UPS* | – | – | **0.59** | **0.25** | **0.28** |
| *Untuned / Unsupervised* | | | | | |
| SumSeg* | – | – | – | 0.438 | 0.455 |
| Gemini | 0.0680 | 0.3300 | 0.1120 | 0.300 | 0.340 |
| Ours | **0.1600** | **0.5300** | 0.2450 | 0.370 | 0.390 |

MiniSeg attains the highest precision (0.3109), its recall is substantially lower (0.1692), suggesting a conservative segmentation strategy that misses many true boundaries. Traditional baselines such as BiLSTM and TopSeg perform poorly across all metrics, with low F1 scores and higher $P_k$ and WD values, reflecting limited ability to capture topic transitions in long, multi-turn meetings. Pretrained embedding-based models (MiniLM and Mpnet) show modest improvements over these baselines but remain clearly behind our method, particularly in recall. Gemini achieves relatively high recall (0.360) but low precision (0.074), leading to over-segmentation and only moderate overall performance.

On the TIAGE dataset, we observe similar performance trends. The UPS (Yang et al., 2025b) model which is a supervised baseline explicitly pre-trained on the TIAGE dataset via utterance-pair modeling showed the best performance across all metrics. While UPS achieves higher absolute segmentation scores ($F_1$: 0.59 vs. 0.245), this gain is primarily due to its task-specific supervision. In contrast, our approach operates in a strictly unsupervised, zero-shot capacity, achieving the highest $F_1$ and recall among untuned methods while attaining the closer $P_k$ (0.370) and WD (0.390). These results demonstrate that our structural inference generalizes across disparate domains—from meeting transcripts to social stories—without any dependency of extensive fine-tuning on the dataset.

**Error analysis:** The error patterns differ across the two datasets and reflect their discourse structure. On QMSum, errors often occur during gradual topical drift, for example when a meeting transitions from an agenda recap into a decision-making phase without an explicit shift in vocabulary; such transitions may not yield sharp cosine-depth peaks, leading to occasional under-segmentation. On TIAGE, errors are more frequently driven by rapid speaker alternation and informal conversational style, where lexical variation across turns can trigger spurious boundaries despite stable topical content. While semantic coherence pruning mitigates many of these cases, distinguishing discourse-level topic change from surface-level conversational dynamics remains challenging.

### 4.1.5 Ablation study: Topic Segmentation pipeline

Table 6 (Left) summarises the contribution of each module to our segmentation pipeline. Precision here reflects boundary reliability, i.e., the proportion of predicted boundaries that align with true topic transitions. The complete model achieves the highest Precision (0.17) and F1 score (0.26), suggesting that each stage of the architecture contributes meaningfully to overall performance. The most significant performance degradation occurs when contrastive pre-training is removed, resulting in a drop in the F1 score from 0.26 to 0.14. This confirms that pre-training is essential for generating the discriminative embeddings required to identify topic transitions. Similarly, omitting semantic pruning results in a 42% decrease in F1 (0.15), indicating that local boundary detection alone is insufficient without a final coherence check to remove spurious segments. The Okta filtering module primarily governs the model's Precision. Removing Okta results in a

sharp decline in Precision (from 0.17 to 0.08) and a corresponding increase in Recall. Although the 'w/o Okta' variant yields lower $P_k$ and WD scores, this reflects a known artefact of distance-based metrics under over-segmentation, where frequent boundary predictions reduce average boundary distance without improving boundary correctness. The substantial drop in F1 confirms that Okta filtering is essential for maintaining boundary reliability. Finally, the Gradient filtering module serves as a secondary noise-reduction step. Removing it increases $P_k$ and WD errors by 0.02, signalling a loss in boundary placement accuracy. When both Okta and Gradient filters are disabled, the system defaults to an aggressive segmentation strategy (Recall: 0.71), which fails to maintain the structural integrity of the topics. Overall, the results support a multi-stage segmentation pipeline with contextual pre-training, boundary filtering, and semantic pruning.

### 4.2 Experiment 2 - Query focused summarization

#### 4.2.1 Datasets:

We evaluate the performance of our topic-guided query-focused summarisation model across three distinct benchmarks, covering narrative storytelling (ODSum-Story (Rao et al., 2024)) to complex, multi-party meeting transcripts (QMSum and ODSum-Meeting (Rao et al., 2024)), all of which provide query-conditioned reference summaries. Further details are added in Appendix A.5.1.

#### 4.2.2 Baselines:

We evaluate the proposed TopicSummRAG against a diverse set of baselines categorized by their learning paradigm and architectural approach:

- Zero-shot

  - LLaMA-3.1-8B, Qwen-3-8B, Gemma-3-12B and Deepseek: Recent state-of-the-art open-source LLMs evaluated in a zero-shot capacity. These models generate summaries directly from the full input (subject to model limits) without retrieval or task-specific fine-tuning.
  - LLaMA-2-13B (Resummarize): A summarization via chapterization strategy proposed by Laskar et al. (2023), where the transcript is divided into fixed-length chunks ("chapters"), each chunk is summarized independently, and a final summary is generated by re-summarizing or rewriting the concatenated chapter summaries. This approach mitigates context length limitations but does not use retrieval or external grounding.
  - Mistral-7B-Instruct-v0.3 (Zero-Shot): Evaluated using the multi-query optimization strategy proposed by Laskar et al. (2024), focusing on the model's ability to synthesize summaries from single-query prompts without additional grounding.

- Supervised Fine-Tuning (SFT)

  - BART-Large (Summ$^N$): A multi-stage, split-then-summarize framework proposed by Zhang et al. (2022). It addresses the context window limitations of standard Transformers by recursively generating coarse summaries before producing a final fine-grained output. This model represents a strong supervised baseline for long-form dialogue.
  - DialogLM: A specialized encoder-decoder model pre-trained on long-range dialogue data. It utilizes a window-based sparse attention mechanism to capture long-distance dependencies in multi-party conversations.
  - Mistral-7B-Instruct-v0.3 (SFT): A fine-tuned variant of the Mistral-7B architecture, trained following the supervised setting described by Laskar et al. (2024), where the model is optimized for instructional adherence in meeting contexts.
    All SFT models are trained using labeled summarization data and therefore benefit from task-specific supervision.

- Retrieval-Augmented Generation (RAG):

- Document Segmentation Matters for RAG (Wang et al., 2025): It proposes PIC (Pseudo-Instruction for Chunking), using document summaries as pseudo-instructions to group semantically coherent sentences via similarity. We used the Llama-3.1-8B model for generating the pseudo-instructions and later for generation as well keeping all other parameters similar to our retrieval settings.

- To isolate the effect of segmentation, we keep retrieval and generation components consistent across methods and vary only the chunking strategy. We conducted our experiments using various retrieval and answer generation models. For retrieval, we experiment with three different algorithms: BM25 (Robertson & Zaragoza, 2009), Sentence-BERT (SBERT) (Reimers & Gurevych, 2019), and Situated Embeddings (Situated) (Wang et al., 2023). Further details are added in Appendix **??**.

  For answer generation, we experimented in conjunction with three state-of-the-art open-source LLMs: LLaMA-3.1-8B[5], Gemma-3[6], and Qwen-3-8B, enabling assessment of our method's robustness across diverse model architectures. Following prior work on chunking mechanisms used in RAG (Gao et al., 2024), we further compared our proposed TopicSummRAG with three different RAG chunking mechanisms: (1) Fixed-size, splitting documents into contiguous spans of predefined length; (2) Sentence, treating each sentence as an independent retrieval unit; and (3) Semantic, segmenting documents based on embedding similarity between adjacent sentences using an off-the-shelf library.

  To ensure the observed gains are a result of our architectural design, we enforce a strict budget parity. For all retrieval strategies—including Fixed-size chunks and Semantic Chunking (Kamradt, 2024)—the total retrieval context is fixed. Specifically, for the Semantic Chunking baseline, we utilized the *all-mpnet-base-v2* embedding model to maintain consistency with our framework. We implemented the breakpoint logic by calculating the cosine distance between adjacent sentence embeddings using a sliding window of 3 sentences. The splitting threshold was set at the 95th percentile of distance gradients, a configuration cited as optimal for high-density dialogue like QMSum. All methods utilize the exact same prompt template and the same backbone LLM (LLaMA-3.1-8B/Gemma-3/Qwen-3-8B) for final generation. By fixing the embedding model, the retrieval budget (token count), and the generation prompt across all experiments, we ensure that TopicSummRAG's performance gains are strictly due to its segmentation and entropy-based adaptive retrieval logic rather than an increased information allowance. Overall, our evaluation controls for model, prompt, and retrieval budget, allowing us to attribute performance differences primarily to the segmentation strategy rather than differences in model or context size.

### 4.2.3 Metrics:

Inspired by Kirstein et al. (2024), we evaluate summary quality using a complementary set of metrics that capture lexical overlap (ROUGE-1,2,L; (Lin, 2004)), semantic alignment (LENS; (Maddela et al., 2023)), BertScore (BS), and referenceless functional utility (BLANC; (Vasilyev et al., 2020)). Together, these metrics provide a holistic assessment of model performance, accounting for diverse error types such as omissions, hallucinations, and coherence issues. Additional details are provided in Appendix A.5.2.

### 4.2.4 Results:

**Performance Against Supervised Baselines:** Table 4 shows that TopicSummRAG is able to close much of the gap between zero-shot methods and supervised fine-tuning on QMSUM. Using LLaMA-3.1-8B for generation, our approach reaches an R1 score of 0.3385, which is very close to BART-Large (0.3403) and DialogLM (0.3402). This suggests that providing better-structured and more coherent context to the model can boost the performance, even without training on labeled data. In contrast, the standard zero-shot LLaMA-3.1-8B setup achieves 0.2693, so the improvement from our method is quite substantial (around 25.7%). This indicates that a large part of the performance gap comes from how context is selected and

---

[5]meta-llama/Llama-3.1-8B-Instruct
[6]google/gemma-3-12b-it

Table 4: Summarization performance on QMSUM across different summarization settings. * indicates results reported directly from the respective papers. (Best model score is in bold and second best is underlined)

| Backbone Model | R1 | R2 | RL | BS |
|---|---|---|---|---|
| *Zero-shot* | | | | |
| LLaMA-3.1-8B | 0.2693 | 0.0625 | 0.1718 | 0.1474 |
| Gemma-3-12B | 0.2017 | 0.0440 | 0.1275 | 0.0598 |
| LLaMA-2-13b chapter* (Laskar et al., 2023) | 0.2901 | 0.0571 | 0.1764 | 0.5549 |
| Mistral-7B-Instruct-v0.3* (Laskar et al., 2024) | 0.2640 | 0.0700 | 0.1700 | 0.5800 |
| Qwen-3-8B | 0.2737 | 0.0673 | 0.1655 | 0.1577 |
| Deepseek | 0.1727 | 0.0234 | 0.1257 | 0.0037 |
| *SFT* | | | | |
| Mistral-7B-Instruct-v0.3* (Laskar et al., 2024) | 0.3000 | 0.1010 | 0.2090 | 0.5990 |
| BART-Large* (Zhang et al., 2022) | **0.3403** | **0.0993** | 0.2900 | – |
| DialogLM* | 0.3402 | 0.0919 | **0.2977** | – |
| *RAG* | | | | |
| LLaMA-3.1-8B (Sentence-wise) | 0.2504 | 0.0560 | 0.1692 | 0.7852 |
| LLaMA-3.1-8B (Fixed Size) | 0.2558 | 0.0529 | 0.1599 | 0.7901 |
| LLaMA-3.1-8B (Doc segmentation (Wang et al., 2025)) | 0.2735 | 0.0622 | 0.1874 | 0.8104 |
| LLaMA-3.1-8B (TopicSummRAG - Ours) | 0.3385 | 0.0977 | 0.2135 | **0.8417** |

Table 5: Summarization performance on ODSUM-Story (Left) and ODSUM-Meeting (Right).

| **ODSUM-Story** | | | | **ODSUM-Meeting** | | |
|---|---|---|---|---|---|---|
| Model | R2 | BS | | Model | R2 | BS |
| BART* (Rao et al., 2024) | **0.109** | 0.844 | | BART* (Rao et al., 2024) | **0.108** | 0.863 |
| Ours | 0.061 | **0.862** | | Ours | 0.087 | **0.867** |

organized, rather than limitations of the model itself. We observe similar patterns on the ODSUM datasets (Table 5). On both ODSUM-Story and ODSUM-Meeting, BART achieves higher ROUGE-2 scores, but our method consistently gives better BERTScore (0.862 vs. 0.844 and 0.867 vs. 0.863, respectively). This suggests that while our summaries may not always match the exact phrasing of the reference as closely, they tend to stay closer in meaning. Overall, these results show that TopicSummRAG generalizes reasonably well across different types of data, without relying on dataset-specific fine-tuning.

On average, a single query requires 30.05 seconds end-to-end, with final answer generation accounting for 28.11 seconds, topic-aware coarse retrieval with entropy-based adaptive selection accounting for 1.93 seconds, and sentence-level fine-grained retrieval within matched regions contributing negligible latency (<0.01 seconds).

**Impact of Topic-Aware Chunking:** To isolate the benefit of our segmentation strategy, we compare TopicSummRAG against fixed-size, sentence-level, and semantic chunking across ODSum-Story, ODSum-Meeting, and QMSum under multiple retrieval strategies (Figure 2). Across all datasets, TopicSummRAG consistently improves query-focused summarization quality, indicating that aligning retrieval units with topical structure yields more relevant and coherent evidence for generation. On ODSum-Story and ODSum-Meeting, TopicSummRAG achieves substantial gains over all baselines across ROUGE, LENS, and BLANC metrics. In particular, improvements in ROUGE-2 and ROUGE-L indicate better coverage of query-relevant content and improved sequence-level coherence, while higher LENS and BLANC scores suggest that retrieved segments form semantically unified and generation-friendly contexts. These gains are consistent across BM25, SBERT, and Situated retrieval, demonstrating that TopicSummRAG complements both lexical and seman-

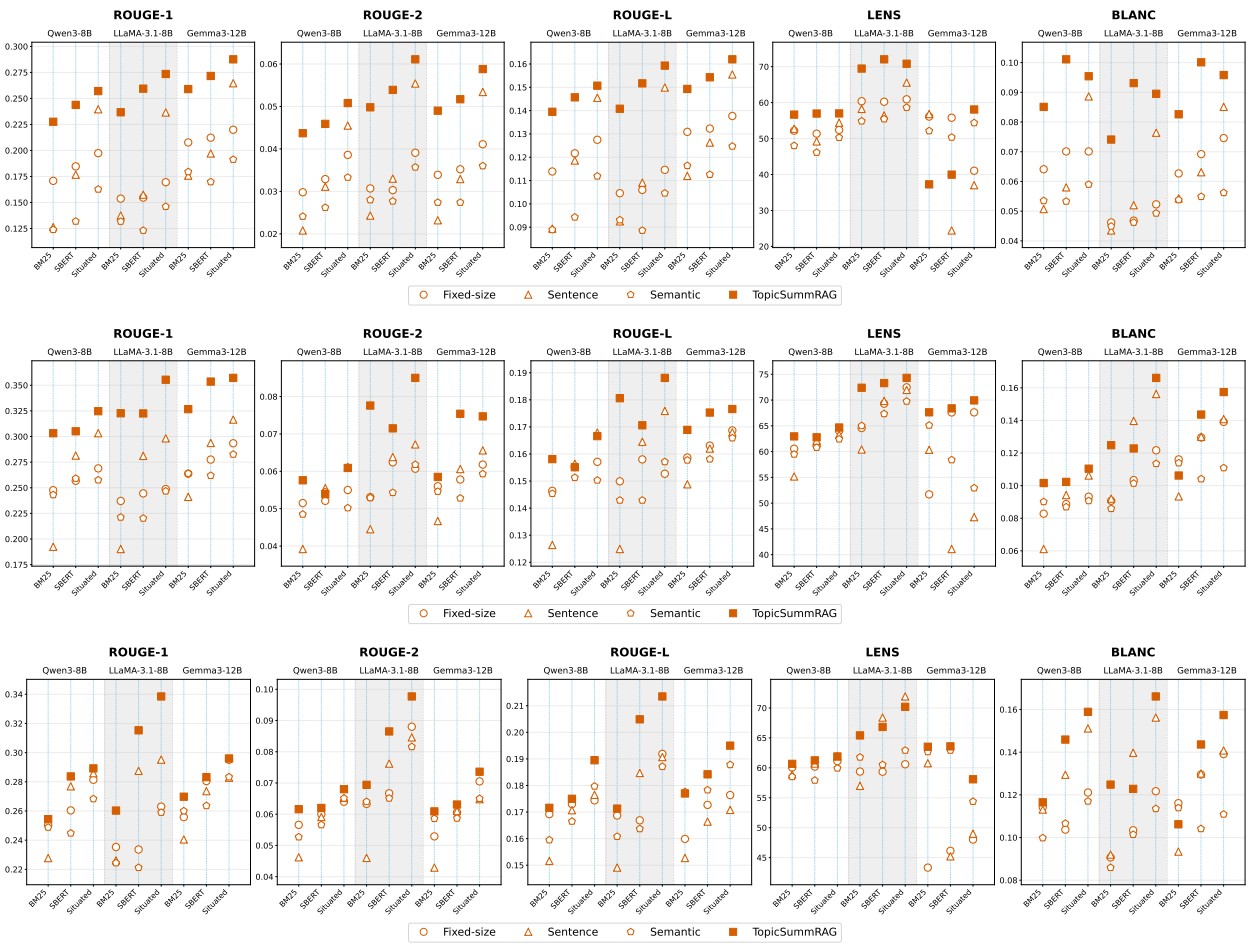

Figure 2: Results on ODSum-Story (top), ODSum-Meeting (middle), and QMSum (bottom) datasets.

tic retrievers. On QMSum, TopicSummRAG similarly outperforms baseline chunking strategies across all evaluated generators. Gains in ROUGE-2 and ROUGE-L reflect improved grounding in query-relevant discourse spans, while higher LENS and BLANC scores confirm increased semantic coherence and factual utility. Importantly, improvements persist across different generator architectures, indicating that the benefits of topic-aware chunking are model-agnostic rather than tied to a specific LLM. A manual audit in Appendix A.11 confirms that TopicSummRAG significantly reduces retrieval and summarization failures compared to these baselines.

**Qualitative evaluation via G-EVAL:** While traditional metrics measures tokens overlap, we also used G-EVAL to assess the quality of the generated summaries. Table 3 reports the G-EVAL scores on top-performing models from each dataset such as ODSum-Story (Gemma-3-12B), ODSum-Meeting (LLaMA-3.1-8B), and QMSum (LLaMA-3.1-8B) (details in Appendix A.8). TopicSummRAG consistently achieves the consistently competitive or best-performing scores across all metrics and retrieval methods, demonstrating that topic-aware chunking produces semantically coherent, informative, and fluent summaries. High G-EVAL scores indicate that summaries are factually accurate, contextually coherent, topically aligned, and easily interpretable by downstream models. On the ODSum datasets, it yields notable gains in Relevance, Coherence, and Factuality, while on QMSum it similarly outperforms baseline chunking strategies, confirming its robustness and generalizability across models and meeting domains. Overall, these results confirm that topic-aware segmentation improves query-focused summarization by providing retrieval units that are both semantically coherent and aligned with the underlying discourse structure, across datasets, retrievers, and generators. To further validate these LLM-based scores, we performed a human-led error analysis using a

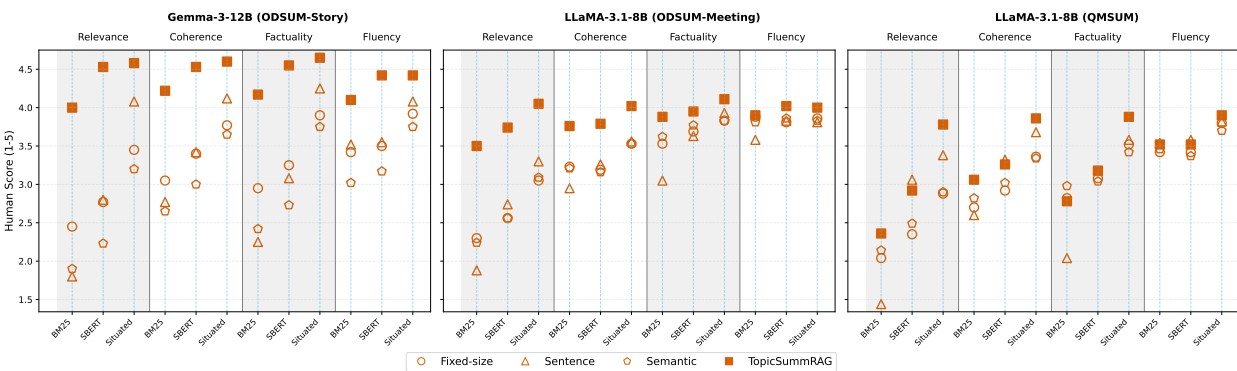

Figure 3: G-EVAL scores

fine-grained taxonomy (RE and SU codes), which reveals that TopicSummRAG mitigates 86.8% of retrieval noise seen in standard RAG (details in Appendix A.11).

**Disentangling Structural and Generative Performance:** The performance gain is qualitatively evidenced by our Case Studies (Appendix A.11, Tables 16, 17, 18). To investigate the causal link between topical segmentation, retrieval accuracy, and final summarization quality, we conducted a granular correlation analysis (detailed in Appendix A.12). Our findings reveal a near-linear relationship between Topic-level F1 and Retrieval F1@5 (Pearson $r = 0.91$), suggesting that the topic purity of segments acts as a support for the retrieval mechanism. Furthermore, the strong correlation between Retrieval F1 and ROUGE-L ($r = 0.95$) confirms that the generative models (e.g., LLaMA-3.1) relies on the topical grounding provided by our hierarchical pipeline. This evidence suggests that the performance gains in TopicSummRAG are not merely stochastic but due to chain of information flow. To evaluate the internal dependencies of the TopicSummRAG pipeline, we analyze the "tripartite flow" from structural segmentation to final generation. We introduce Topic Purity (0.703) as an intermediate metric to quantify how closely our predicted boundaries align with ground-truth discourse units. Our analysis reveals that high topic purity directly minimizes retrieval noise (RE-03 reduced from 86.84% to 5.26%) and ensures superior Query Coverage.Statistical validation confirms a near-linear correlation between Retrieval F1@5 and ROUGE-L (Pearson $r = 0.9516, p < 0.001$), suggesting that structural alignment is the primary bottleneck for long-document QFS. When this alignment is preserved, generative fidelity follows as a mechanistic consequence. A detailed breakdown of these intermediate metrics, including formal correlation protocols and qualitative case studies (e.g., Barry Hughes and Auditor General examples), is provided in Appendix A.12.

**Error analysis:** We analyse cases where TopicSummRAG underperforms baseline chunking on QMSum. For highly specific queries targeting a single action item or decision, topic-level chunks may include additional surrounding discussion that is topically related but not strictly necessary to answer the query, reducing relevance under dense retrieval. Under BM25 retrieval, summarizing multiple related utterances within a topic can occasionally lead to mild over-generalization, where fine-grained details are merged, which is penalized by G-EVAL factuality scores. Notably, these errors do not affect fluency or coherence, indicating that failures stem from context aggregation rather than hallucination. A more detailed error analysis and significant t-test is added in Appendix A.11 and A.10. To rigorously categorize these behaviors, we introduce an extended error taxonomy in Appendix A.11. Our human evaluation across QMSum and ODSum shows that TopicSummRAG achieves a 55% relative reduction in evidence omission (SU-01) compared to fixed-size chunking, directly supporting our claim of superior factual grounding.

### 4.2.5 Ablation Study: TopicSummRAG pipeline

Table 6 (Right) evaluates different retrieval configurations. The Sentence RAG baseline achieves moderate F1 (0.218) but suffers from noisy retrieval, including many irrelevant sentences. The TopicSumm module alone yields a very high recall (0.866), effectively capturing relevant topical chunks for the query. Intro-

Table 6: Combined ablation studies: (Left) Segmentation pipeline; (Right) TopicSummRAG results on QMSum. (*E*: Entropy, *summ*: summarization). *Note:* F1 is computed as the macro-average of per-document F1 scores

| Method | P (↑) | R (↑) | F1 (↑) | $P_k$ (↓) | WD (↓) |
|---|---|---|---|---|---|
| Ours | **0.17** | 0.56 | **0.26** | 0.36 | 0.38 |
| w/o contrastive | 0.12 | 0.56 | 0.14 | 0.39 | 0.38 |
| w/o Okta | 0.08 | **0.59** | 0.13 | **0.32** | **0.33** |
| w/o Grad | 0.10 | 0.55 | 0.15 | 0.38 | 0.40 |
| w/o Okta & Grad | 0.14 | 0.71 | 0.21 | 0.37 | 0.39 |
| w/o semantic pruning | 0.11 | 0.44 | 0.15 | 0.36 | 0.38 |

| Method | P (↑) | R (↑) | F1 (↑) |
|---|---|---|---|
| Sentence RAG | 0.17 | 0.36 | 0.21 |
| TopicSumm | 0.02 | **0.86** | 0.04 |
| TopicSumm + E | 0.07 | 0.78 | 0.12 |
| TopicSumm + E → RAG | **0.39** | 0.26 | **0.27** |
| w/o chunk-level summ | 0.28 | 0.25 | 0.26 |

ducing entropy-based adaptive selection further reduces retrieval noise by pruning weakly relevant topical segments, improving precision while preserving relevant content. Combining TopicSumm, entropy filtering, and RAG retrieval produces the best-balanced performance, achieving the highest precision (0.392) and F1 score (0.272). This progression demonstrates how each stage incrementally cleans the retrieval output, from broad coverage to focused, concise, and query-relevant results. Removing chunk-level summarisation forces coarse retrieval to operate over raw segment embeddings, which often mix multiple subtopics and dilute topical signals, leading to noisier retrieval and reduced precision. Although recall remains comparable due to broader coverage, the resulting context is noisier, leading to reduced precision and F1. This result highlights that topic segmentation alone is insufficient: high recall must be complemented by adaptive selection and fine-grained retrieval to achieve precise, query-focused evidence.

The ablations demonstrate that TopicSummRAG's gains arise from the joint interaction of contextual segmentation, adaptive filtering, and fine-grained retrieval, with each component contributing to either structural coherence or retrieval precision. To further assess robustness beyond component-level ablations, we report additional sensitivity analyses in Appendix A.6, examining the effects of retrieval granularity, temperature scaling, contextual window size, and candidate set size; these results confirm that TopicSummRAG's gains are stable across a broad range of hyperparameter choices.

### 4.2.6 Hyperparameters

We finetune the `unsup-simcse-roberta-base` encoder using the proposed context-aware contrastive learning objective and generate unit-level embeddings from the finetuned model. Embeddings are computed with a maximum token length of 256 and chunked inference (batch size 50), ensuring stable GPU utilisation across documents of varying lengths. For topical boundary detection, semantic depth is computed using a sliding contextual window of size $W$. We set $W = 3$ for short dialogues and $W = 5$ for longer documents, reflecting a trade-off between sensitivity to local semantic transitions and robustness to conversational noise. To avoid repeated detections of the same topical transition, we enforce a minimum inter-peak distance of 3 units for short documents and 7 units for longer meetings, which prevents over-fragmentation while preserving genuine boundary candidates. We incorporate lightweight structural cues via a feature augmentation module consisting of a projection layer (32 or 64 dimensions) followed by a transformer mixer with 1–2 layers and 2–4 attention heads. This design introduces limited contextual interaction to stabilise depth estimates without adding substantial computational overhead. Larger configurations did not yield measurable improvements in preliminary experiments. Candidate boundary selection relies on two complementary detectors: Otsu thresholding, which provides a global, distribution-based cutoff over depth values, and gradient-based detection, which identifies sharp local semantic changes. A boundary is retained only if both detectors agree within a ±3-unit window, accommodating minor positional offsets introduced by contextual windowing. Following candidate selection, semantic coherence pruning regulates final segmentation granularity. Boundaries whose resulting segments exhibit cross-segment coherence above 0.99 are removed, as such values indicate near-identical semantic distributions across the split. A minimum segment length of three units is enforced to eliminate trivial partitions.

For topic-guided retrieval, each segment is summarised using Qwen3-8B and query–segment similarities are computed using SBERT. To correct for the compressed numeric range of cosine similarity scores, we apply temperature scaling with $\tau = 0.3$ before softmax normalisation, following standard practice in contrastive

retrieval models. We set $L = 5$ for QMSum and $L = 10$ for ODSum, reflecting differences in topical breadth across datasets; sensitivity to $L$ is analysed in Appendix A.6. The number of retained segments $k$ is determined adaptively based on the dominance of the relevance distribution: $k = 1$ when the maximum softmax probability exceeds 0.6, $k = 2$ when it lies between 0.4 and 0.6, and $k = L$ otherwise. Entropy is computed to characterise uncertainty in the similarity distribution but is not used directly to determine $k$. In the fine-grained retrieval stage, selected segments are decomposed into sentences, and the top sentences are retrieved using dense similarity matching. This value balances evidence coverage with prompt length constraints and is held fixed across datasets.

## 5 Conclusion

We introduced TopicSummRAG, a framework that addresses the limitations of naïve chunking in retrieval-augmented generation for long, multi-topic documents. The approach is built on a topic segmentation pipeline that partitions documents into semantically coherent discourse units, aligning retrieval with latent topical structure rather than arbitrary textual boundaries. Evaluations on QMSum and TIAGE show that this segmentation model achieves more accurate boundary detection than strong baselines, particularly for long, multi-turn conversations. Building on these segments, TopicSummRAG applies segment-level summarization and dominance-based adaptive retrieval to regulate retrieval granularity. This design improves retrieval precision for focused queries while preserving coverage for broader information needs. Across multiple retrievers and LLMs, TopicSummRAG consistently improves factual grounding, coherence, and summary quality. These results highlight the importance of topic-aware indexing for robust long-document RAG, though highly localized queries may occasionally benefit from finer-grained retrieval units.

## 6 Limitation

TopicSummRAG introduces additional computation due to segment-level summary generation prior to retrieval, although this overhead is lower than approaches that rely on full LLM-based segmentation or repeated long-context inference (Lee et al., 2025; Duarte et al., 2024; McCarthy et al., 2023). Additionally, because retrieval operates over coherent topical regions, highly localized queries may occasionally include limited surrounding context, reflecting a trade-off between discourse coherence and minimal evidence selection.

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

# A  Appendix

## A.1  Algorithm for Contextually Coherent Topic Segmentation (CCDS)

## A.2  Example of context-aware triplet construction on QMSum meeting

Figure 4 shows the example of creating triplets for contrastive model training.

## A.3  Detailed pipeline

Figure 5 shows the detailed step-by-step topic segmentation pipeline and TopicSummRAG using an example of a long story.

## A.4  Experiment 1- Topic Segmentation

### A.4.1  Datasets

We evaluate our topic segmentation model on two distinct benchmarks that capture different aspects of conversational dynamics: open-domain social chitchat (TIAGE (Xie et al., 2021)) and multi-party formal meetings (QMSum (Zhong et al., 2021)).

TIAGE (Xie et al., 2021) is a specialized benchmark constructed by augmenting the PersonaChat dataset (Zhang et al., 2018) with human-annotated topic shifts. Unlike general dialogue datasets, TIAGE focuses

---

**Algorithm 1** Contextually Coherent Topic Segmentation (CCDS)

---

**Require:** Transcript $T = \{t_1, \ldots, t_N\}$, dialogue embedder $\mathcal{E}$, window size $W$, minimum boundary index $m_b$, feature augmenter $\mathcal{A}$, hyperparameters $\Theta$

**Ensure:** Predicted topic boundaries $\mathcal{B}$

1:  $E \leftarrow [\, e_i = \mathcal{E}(t_i) \,]_{i=1}^N$        ▷ contextual embeddings
2:  $E' \leftarrow \mathcal{A}(E)$        ▷ feature augmentation
3:  **for** $i \leftarrow 1$ **to** $N$ **do**
4:      $L_i \leftarrow \mathrm{mean}\left(E'_{\max(1,i-W):i-1}\right)$        ▷ left context
5:      $R_i \leftarrow \mathrm{mean}\left(E'_{i+1:\min(N,i+W)}\right)$        ▷ right context
6:      $d_i \leftarrow \mathrm{cosine\_depth}(e_i, L_i, R_i)$
7:  **end for**
8:  $D \leftarrow \{d_1, \ldots, d_N\}$        ▷ depth signal
9:  $\mathrm{peaks} \leftarrow \mathrm{argrelextrema}(D, \mathrm{order} = \lfloor W/2 \rfloor)$
10: $\tau \leftarrow \mathrm{OtsuThreshold}(D)$
11: $\mathcal{C}_1 \leftarrow \{p \in \mathrm{peaks} \mid d_p \geq \tau\}$
12: $\mathcal{G} \leftarrow \mathrm{gradient\_candidates}(D)$
13: $\mathcal{C}_2 \leftarrow \{p \in \mathcal{C}_1 \mid \exists g \in \mathcal{G}, |p - g| \leq W\}$
14: Compute homogeneity $H \leftarrow \sigma(D)/(\mu(D) + \epsilon)$
15: Adapt pruning thresholds in $\Theta$ using $H$
16: $\mathcal{B} \leftarrow \pi(\mathcal{C}_2, D, E; \Theta)$        ▷ semantic coherence pruning
17: $\mathcal{B} \leftarrow \{b \in \mathcal{B} \mid b \geq m_b\}$
18: Ensure boundary at index $N - 1$
19: **return** sorted unique $\mathcal{B}$

---

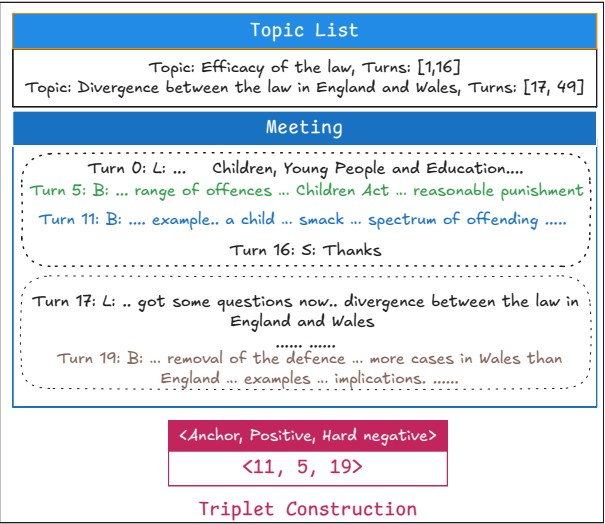

Figure 4: Topic 1 (Turns 1–16) covers efficacy of the law; Topic 2 (Turns 17–49) covers divergence between England and Wales. Turn 11 (Anchor) is paired with Turn 5 (Positive, same topic) and Turn 19 (Hard Negative, different topic but semantically similar). Gold topic spans enable mining of positives and hard negatives that promote embeddings separating semantically overlapping yet topically distinct turns.

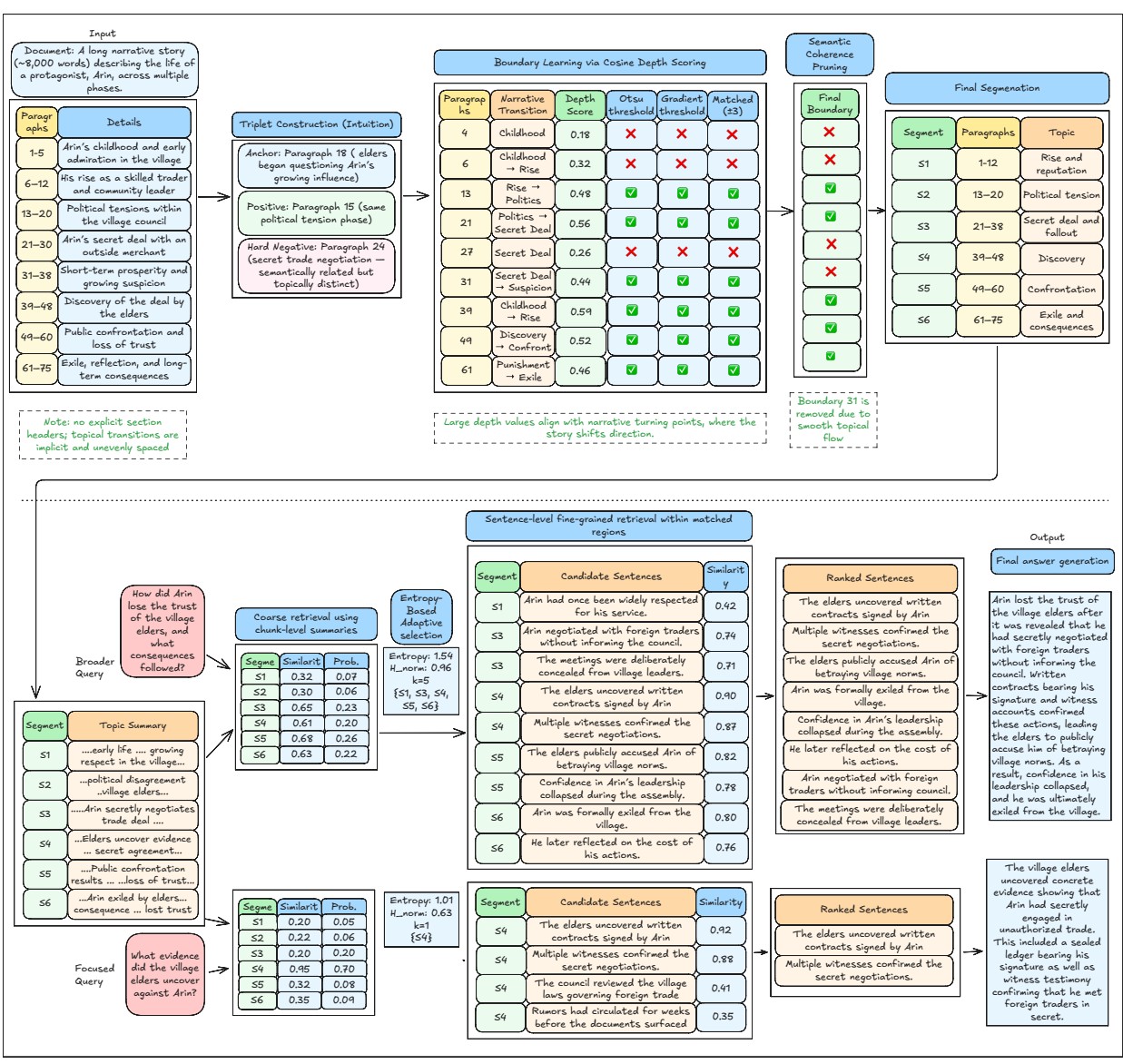

Figure 5: Pipeline of TopicSummRAG (below) and Topic Segmentation (above)

explicitly on the transition points where a conversation fluently moves from one persona-based topic to another. The dataset contains 7,861 gold-standard topic-shift annotations across 500 multi-turn dialogues. We utilise the human-annotated portion of the dataset, which provides a high-quality ground truth for evaluating a model's ability to detect natural conversational flow and proactive topic transitions. QMSum (Zhong et al., 2021) is a large-scale, multi-domain benchmark originally designed for query-based meeting summarisation. It consists of 232 meetings with 1,808 query-summary pairs across three domains: Product (AMI), Academic (ICSI), and Committee discussions. The detailed statistics for both datasets are summarised in Table 2. Although both datasets are conversational, they differ substantially in discourse structure and topical granularity, allowing us to evaluate robustness across both informal and highly structured settings.

### A.4.2 Metrics

**Classification Metrics** The standard metrics include Precision, Recall, and the F1-score. In the context of text segmentation, these are calculated by treating each potential boundary position (e.g., sentence boundary) as a binary classification point. Precision denotes the proportion of predicted boundaries that align with the ground truth; Recall measures the proportion of actual boundaries correctly identified by the model; and the F1-score serves as the harmonic mean of the two to provide a balanced assessment of overall accuracy.

**Distance-Based Metrics** Standard classification metrics are often insufficient for segmentation because they do not account for "near misses" (e.g., a boundary placed one sentence away from the truth). To address this, we utilize $P_k$ and WindowDiff, which are the de facto standards in the topic segmentation field (Retkowski & Waibel, 2024; Badjatiya et al., 2018). The $P_k$ metric (Beeferman et al., 1999) measures the probability that two sentences chosen at random at a distance $k$ are incorrectly classified as being in the same or different segments. A lower $P_k$ score indicates higher segmentation quality. WindowDiff (Pevzner & Hearst, 2002) improves upon $P_k$ by moving a sliding window of size $k$ across the text and comparing the count of boundaries within that window. These metrics that both missing and extraneous boundaries are penalized more equitably, providing a robust measure of the model's structural performance.

### A.5 Experiment 2 - Query focused summarization

### A.5.1 Datasets

We evaluate the performance of our topic-guided query-focused summarisation model across three distinct benchmarks. These datasets are selected to represent varied communicative contexts, ranging from narrative storytelling (ODSum-Story (Rao et al., 2024)) to complex, multi-party meeting transcripts (QMSum and ODSum-Meeting (Rao et al., 2024)).

The ODSum benchmark (Rao et al., 2024) is a recently introduced suite designed for open-domain multi-document summarisation (ODMDS). It consists of two sub-datasets: ODSum-Meeting and ODSum-Story. ODSum-Story is adapted from QMSum, focuses on meeting transcripts, but is structured in an open-domain format where relevant information must be identified and consolidated from an interdependent document index. ODSum-Story is adapted from the SQuALITY dataset (Wang et al., 2022), consisting of public-domain short stories paired with complex, non-factoid queries. Unlike the conversational datasets, ODSum-Story provides narrative inputs, requiring the model to demonstrate a deep understanding of themes and the ability to extract abstractive, query-tailored summaries from dense literary prose. Table **??** provides the comprehensive statistics for each dataset.

### A.5.2 Metrics

Inspired by (Kirstein et al., 2024), we evaluate the quality of the generated summaries using a complementary suite of metrics that assess lexical overlap, semantic alignment, and referenceless functional utility. By combining these different approaches, we provide a holistic view of model performance that accounts for various types of summary errors, including omissions, hallucinations, and a lack of coherence.

Table 7: Sensitivity of TopicSummRAG to the candidate set size $L$ on QMSum.

| $L$ | Precision ($\uparrow$) | Recall ($\uparrow$) | F1 ($\uparrow$) |
|---|---|---|---|
| 3 | 0.386 | 0.259 | 0.270 |
| 5 | 0.392 | 0.267 | 0.272 |
| 8 | 0.389 | 0.271 | 0.271 |
| 10 | 0.387 | 0.273 | 0.270 |

We report the ROUGE scores (Lin, 2004), which remain the standard for measuring the lexical similarity between candidate and reference summaries. ROUGE-1 and ROUGE-2 are used to measure the overlap of unigrams and bigrams, respectively, serving as a proxy for content coverage and informational recall; and ROUGE-L to measure the Longest Common Subsequence (LCS), which captures sentence-level structure and fluency more effectively than individual n-gram matches. While ROUGE is effective for lexical consistency, it often fails to capture semantic paraphrasing. Therefore, we utilise LENS (Maddela et al., 2023). LENS is a supervised metric trained on human judgments that uses a cross-encoder architecture to rank summaries based on their faithfulness and readability. Unlike string-match metrics, LENS can better identify structural disorganisation and subtle semantic deviations that human evaluators typically penalise. To assess the summary's ability to convey the core information of the source document without relying on potentially biased human references, we used BLANC (Vasilyev et al., 2020). BLANC is a referenceless metric that measures the functional quality of a summary by evaluating how much it helps a pre-trained language model perform a masked language modelling (MLM) task on the source document. A high BLANC score indicates that the summary contains sufficient information to help the model "fill in the blanks" of the source text, serving as a strong proxy for document-level information retention.

### A.6   Additional Analysis

#### A.6.1   Retrieval Candidate Size Sensitivity

Table 7 shows that TopicSummRAG exhibits stable performance across a broad range of candidate set sizes. Increasing $L$ slightly improves recall by admitting more candidate segments, while dominance-based adaptive selection prevents the accumulation of irrelevant context for focused queries. As a result, overall F1 remains largely invariant, indicating that retrieval performance is not sensitive to the exact choice of $L$.

#### A.6.2   Hyperparameter Sensitivity and Trade-off Analyses

We further analyze the robustness of TopicSummRAG with respect to retrieval granularity and key hyper-parameters. These analyses complement the ablation study in Section 4.2.5 and illustrate how adaptive retrieval balances precision and recall across different operating regimes.

**Precision–Recall Trade-off under Adaptive Retrieval**   Figure 6 (top-left) illustrates the precision–recall trade-off for different retrieval strategies corresponding to the ablation settings in Table 6. Fixed retrieval granularities (e.g., $k=1$ and $k=L$) exhibit the expected trade-off between precision and recall. In contrast, adaptive-$k$ lies on the Pareto frontier, while retaining substantially more recall than single-segment retrieval. This confirms that dominance-based adaptive selection effectively balances coverage and relevance, and outperforms any fixed choice of $k$.

**Temperature Sensitivity Analysis**   Figure 6 (top-right) reports performance variation as a function of the softmax temperature $\tau$ used for relevance normalization. Performance varies smoothly across the tested range ($\tau \in [0.2, 0.4]$), with $\tau=0.3$ providing a stable operating point. This supports our interpretation of temperature scaling as a mechanism for correcting cosine similarity scale mismatch rather than as a parameter that requires any fine-tuning.

**Dominance Threshold Sensitivity**   We study how sensitive the adaptive retrieval mechanism is to the choice of dominance thresholds $(t_1, t_2)$, which control how $p_{\max}$ is mapped to the number of retrieved segments $k$. Instead of testing only a few settings, we vary both thresholds over a grid spanning $t_1 \in [0.3, 0.7]$ and $t_2 \in [0.2, 0.6]$, while keeping the rest of the pipeline fixed. As shown in Figure 6 (bottom-right), performance changes smoothly across this range and does not show any sharp peak. Instead, a wide range of threshold values gives similar performance, indicating that the method is not sensitive to the exact choice of $(t_1, t_2)$. In general, higher thresholds are more selective and may miss some useful context, while lower thresholds include more segments and may introduce some noise. However, these effects are gradual, and no single setting is clearly better than the others. This is consistent with the candidate set size analysis (Table 7), where $L$ controls how much context is available and the thresholds decide how much of it is used. Together, they help balance coverage and noise. Combined with the precision–recall results, which show that adaptive retrieval performs better than fixed $k$, this suggests that dominance-based selection is a simple and robust approach that does not require careful tuning.

**Context Window Size Sensitivity**   Figure 6 (bottom-right) evaluates the effect of the contextual window size $W$ used for cosine-depth boundary detection. Moderate window sizes ($W{=}3$–5) yield consistent segmentation performance, while very small or very large windows slightly degrade precision or recall. This trend reflects the trade-off between sensitivity to local transitions and robustness to conversational noise, and indicates that TopicSummRAG is not overly sensitive to window size selection.

Together with the candidate set size analysis in Table 7, these results show that TopicSummRAG is robust to moderate variations in retrieval and segmentation hyperparameters.

## A.7   Query-focused summarization prompt

To generate concise, query-focused summaries from meeting transcripts, we designed a structured prompt (Table 8) that guides the model to extract only relevant content. This prompt ensures that the generated summary remains under 150 words and adheres strictly to the query, producing outputs in a standard JSON format. Such prompting helps maintain consistency and reproducibility in evaluation, and can be directly used for zero-shot summarization with LLMs.

Table 8: Prompt for summary generation

| |
|---|
| ***Prompt*** |
| You are an expert in query-focused summary from meetings. Given a query and relevant spans from the meeting, your task is to identify query-relevant content from the given span and generate a query-focused summary. |
| ***Guidelines*** |
| 1. Only include content relevant to the query. 
 2. Keep the query-focused summary under 150 words. 
 3. Output Format: Return a single JSON object with the key `"query_focused_summary"`. Do not include any additional text or explanations. |
| ***Input*** |
| Query: 
 Relevant Span: |

## A.8   G-EVAL summary evaluation

To systematically assess the quality of a query-focused summary, we designed a structured evaluation prompt shown in Table 9. The prompt guides the model to rate a given summary on multiple metrics—relevance, coherence, factuality, and fluency—by comparing it to the story and query. We used *Gemini-2.0-Flash* as the underlying LLM in zero-shot settings.

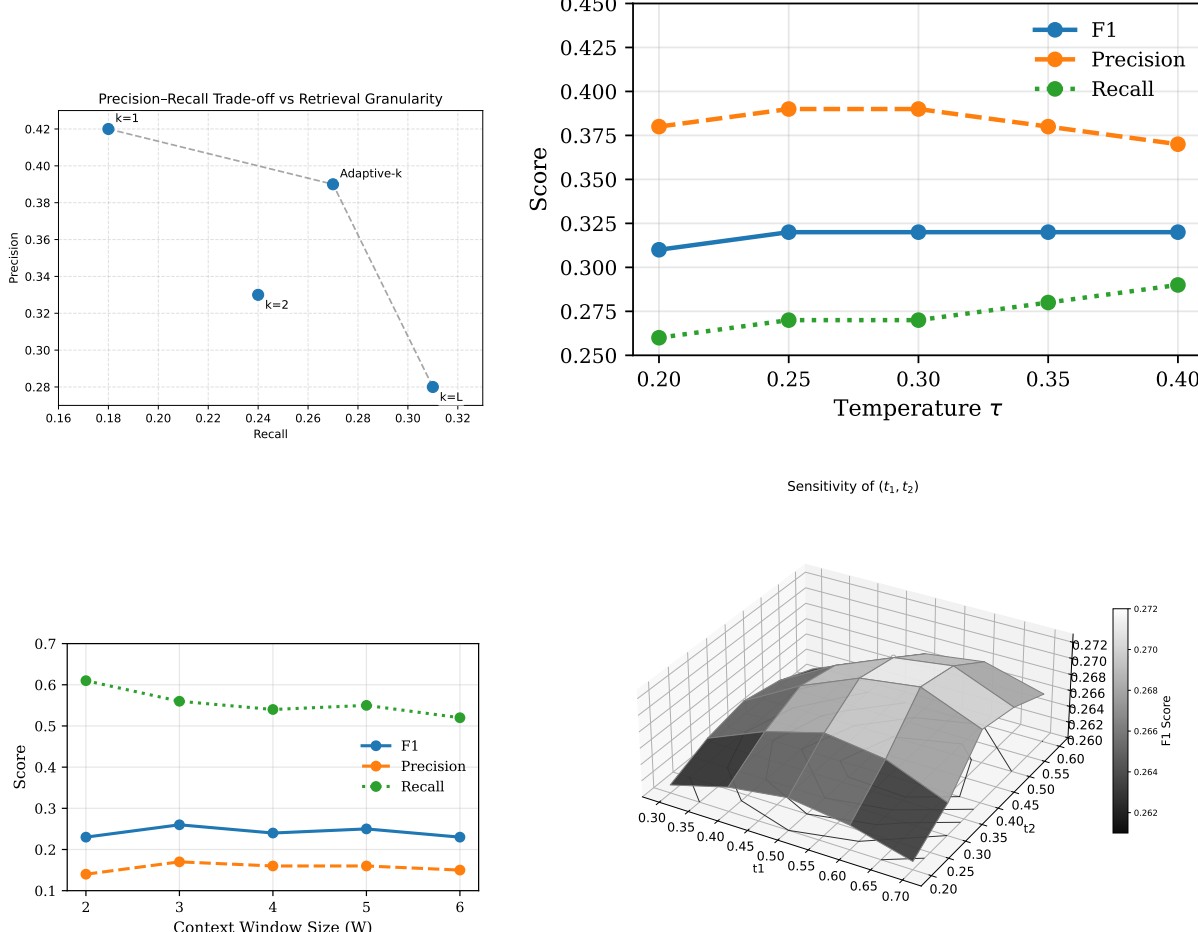

Figure 6: Additional analyses. (top-left) Precision–recall trade-off for fixed and adaptive retrieval strategies. (top-right) Sensitivity to softmax temperature $\tau$. (bottom-left) Sensitivity to contextual window size $W$. (bottom-right) Sensitivity to dominance thresholds $(t_1, t_2)$ showing adaptive retrieval.

## A.9  ODSum-Story Results

The ODSum-story dataset consists of 4 reference summaries associated for each query, hence, we reported the min-max-mean scores among them in Table 10.

## A.10  Significant t-test: Per-Sample ROUGE-L Analysis and Statistical Significance

To quantify the improvement of Situated-TopicSummRAG over baseline, we conducted a paired t-test on ROUGE-L scores computed for each document in our evaluation set.

The paired t-test results are as follows:

- t-statistic: 14.8245

- p-value: $5.38 \times 10^{-36}$

- Mean $\Delta$ ROUGE-L (Situated-TopicSummRAG – baseline): 0.0652

- 95% Confidence Interval: [0.0565, 0.0738]

Table 9: Prompt for G-EVAL

| |
|---|
| ***Task Description*** |
| You are an expert evaluator for query-focused summarization tasks. Your job is to assess the quality of a query-focused summary against a given story and query using multiple evaluation criteria. The evaluation involves rating the summary on relevance, coherence, factuality, and fluency. |
| ***Metric Definition*** |
| The definitions of the evaluation criteria are: |
| **Relevance:** How well the summary addresses the information need expressed in the query. |
| **Coherence:** The logical flow, structure, and clarity of the summary. |
| **Factuality:** The correctness and accuracy of the information presented in the summary compared to the story. |
| **Fluency:** The grammaticality, readability, and overall linguistic quality of the summary. |
| ***Guidelines*** |
| 1. Read the query-focused summary carefully and identify its main topic and key points. |
| 2. Compare the summary to the story and check whether it covers the relevant content clearly and logically. |
| 3. Evaluate the summary on each of the four criteria (relevance, coherence, factuality, fluency) on a scale from 1 (lowest) to 5 (highest). |
| 4. Return the results in JSON format with keys "relevance", "coherence", "factuality", and "fluency". |
| ***Input*** |
| Query: |
| Story: |
| Summary: |
| ***Output*** |
| {"relevance": X, "coherence": X, "factuality": X, "fluency": X} |

The extremely low p-value indicates that the improvement in ROUGE-L is highly significant, confirming that Situated-TopicSummRAG consistently outperforms the baseline in capturing query-focused summary quality. Figure 7 shows the distribution of per-document improvements ($\Delta$ ROUGE-L = Situated-TopicSummRAG - baseline). The majority of documents exhibit positive gains, indicating consistent improvements across the dataset. Overall, both the statistical testing and per-sample analysis demonstrate that Situated-TopicSummRAG provides significant and consistent improvements in query-focused meeting summarization compared to a baseline.

### A.11 Human Evaluation: Error Analysis

To validate the core claim of factual grounding, we conducted a manual audit on a sampled subset of the QMSum and ODSUM datasets. Three independent annotators evaluated the generated summaries by identifying error types using a predefined taxonomy, which serves as a proxy for factual consistency and grounding.

We randomly sampled 100 examples from each dataset. For each example, annotators were provided with: (i) the input query, (ii) the retrieved evidence spans, and (iii) summaries generated by different baselines. Annotators were not shown the model identity (blind evaluation) to avoid bias. We measure inter-annotator agreement using Cohen's $\kappa$ over the assigned error labels, obtaining $\kappa = 0.69$, indicating substantial agreement among annotators.

Table 10: Summarization performance on ODSum-Story dataset. *Bold numbers indicate the best performance within each retriever group.*

| Retrieval | Chunking | ROUGE-1 | | | ROUGE-2 | | | ROUGE-L | | |
|---|---|---|---|---|---|---|---|---|---|---|
| | | Min | Max | Avg | Min | Max | Avg | Min | Max | Avg |
| | | | | *LLaMA-3.1-8B* | | | | | | |
| BM25 | Fixed-size | 0.1211 | 0.1889 | 0.1537 | 0.0174 | 0.0462 | 0.0307 | 0.0827 | 0.1283 | 0.1046 |
| | Sentence | 0.1072 | 0.1696 | 0.1375 | 0.0122 | 0.0386 | 0.0243 | 0.0730 | 0.1140 | 0.0925 |
| | Semantic | 0.1027 | 0.1620 | 0.1318 | 0.0143 | 0.0432 | 0.0280 | 0.0731 | 0.1156 | 0.0931 |
| | **TopicSummRAG** | 0.1978 | 0.2776 | **0.2367** | 0.0315 | 0.0701 | **0.0498** | 0.1172 | 0.1667 | **0.1408** |
| SBERT | Fixed-size | 0.1215 | 0.1906 | 0.1547 | 0.0158 | 0.0474 | 0.0303 | 0.0834 | 0.1308 | 0.1060 |
| | Sentence | 0.1233 | 0.1934 | 0.1575 | 0.0180 | 0.0502 | 0.0330 | 0.0861 | 0.1351 | 0.1091 |
| | Semantic | 0.0930 | 0.1561 | 0.1231 | 0.0138 | 0.0437 | 0.0277 | 0.0685 | 0.1114 | 0.0886 |
| | **TopicSummRAG** | 0.2198 | 0.3014 | **0.2594** | 0.0346 | 0.0752 | **0.0539** | 0.1274 | 0.1798 | **0.1517** |
| Situated | Fixed-size | 0.1352 | 0.2082 | 0.1695 | 0.0225 | 0.0593 | 0.0391 | 0.0913 | 0.1418 | 0.1146 |
| | Sentence | 0.1929 | 0.2807 | 0.2365 | 0.0347 | 0.0786 | 0.0554 | 0.1223 | 0.1804 | 0.1499 |
| | Semantic | 0.1140 | 0.1816 | 0.1461 | 0.0198 | 0.0540 | 0.0357 | 0.0809 | 0.1319 | 0.1046 |
| | **TopicSummRAG** | 0.2322 | 0.3172 | **0.2735** | 0.0389 | 0.0848 | **0.0611** | 0.1349 | 0.1864 | **0.1593** |
| | | | | *Qwen3-8B* | | | | | | |
| BM25 | Fixed-size | 0.1387 | 0.2048 | 0.1708 | 0.0168 | 0.0444 | 0.0298 | 0.0934 | 0.1367 | 0.1139 |
| | Sentence | 0.0989 | 0.1552 | 0.1263 | 0.0102 | 0.0329 | 0.0208 | 0.0705 | 0.1101 | 0.0894 |
| | Semantic | 0.0969 | 0.1519 | 0.1239 | 0.0131 | 0.0367 | 0.0241 | 0.0704 | 0.1094 | 0.0889 |
| | **TopicSummRAG** | 0.1913 | 0.2646 | **0.2276** | 0.0277 | 0.0618 | **0.0437** | 0.1172 | 0.1631 | **0.1395** |
| SBERT | Fixed-size | 0.1495 | 0.2212 | 0.1847 | 0.0191 | 0.0481 | 0.0329 | 0.1000 | 0.1466 | 0.1217 |
| | Sentence | 0.1428 | 0.2136 | 0.1767 | 0.0174 | 0.0465 | 0.0311 | 0.0963 | 0.1428 | 0.1186 |
| | Semantic | 0.1026 | 0.1630 | 0.1319 | 0.0147 | 0.0390 | 0.0262 | 0.0741 | 0.1174 | 0.0943 |
| | **TopicSummRAG** | 0.2074 | 0.2829 | **0.2438** | 0.0296 | 0.0643 | **0.0459** | 0.1241 | 0.1700 | **0.1457** |
| Situated | Fixed-size | 0.1624 | 0.2358 | 0.1974 | 0.0225 | 0.0568 | 0.0386 | 0.1042 | 0.1533 | 0.1275 |
| | Sentence | 0.2010 | 0.2790 | 0.2396 | 0.0288 | 0.0640 | 0.0455 | 0.1220 | 0.1705 | 0.1455 |
| | Semantic | 0.1290 | 0.1984 | 0.1627 | 0.0185 | 0.0505 | 0.0333 | 0.0882 | 0.1365 | 0.1119 |
| | **TopicSummRAG** | 0.2172 | 0.2972 | **0.2571** | 0.0324 | 0.0705 | **0.0508** | 0.1277 | 0.1749 | **0.1507** |
| | | | | *Gemma3-12B* | | | | | | |
| BM25 | Fixed-size | 0.1710 | 0.2455 | 0.2077 | 0.0195 | 0.0506 | 0.0339 | 0.1088 | 0.1554 | 0.1309 |
| | Sentence | 0.1436 | 0.2096 | 0.1758 | 0.0121 | 0.0360 | 0.0232 | 0.0923 | 0.1338 | 0.1120 |
| | Semantic | 0.1455 | 0.2151 | 0.1795 | 0.0144 | 0.0413 | 0.0274 | 0.0959 | 0.1382 | 0.1164 |
| | **TopicSummRAG** | 0.2216 | 0.2979 | **0.2590** | 0.0319 | 0.0676 | **0.0490** | 0.1284 | 0.1721 | **0.1493** |
| SBERT | Fixed-size | 0.1747 | 0.2513 | 0.2123 | 0.0197 | 0.0518 | 0.0352 | 0.1094 | 0.1574 | 0.1323 |
| | Sentence | 0.1602 | 0.2354 | 0.1971 | 0.0190 | 0.0485 | 0.0329 | 0.1050 | 0.1497 | 0.1263 |
| | Semantic | 0.1340 | 0.2066 | 0.1698 | 0.0149 | 0.0414 | 0.0274 | 0.0908 | 0.1352 | 0.1126 |
| | **TopicSummRAG** | 0.2324 | 0.3107 | **0.2716** | 0.0337 | 0.0706 | **0.0517** | 0.1322 | 0.1779 | **0.1543** |
| Situated | Fixed-size | 0.1810 | 0.2611 | 0.2199 | 0.0245 | 0.0605 | 0.0411 | 0.1139 | 0.1635 | 0.1377 |
| | Sentence | 0.2258 | 0.3036 | 0.2646 | 0.0345 | 0.0740 | 0.0534 | 0.1312 | 0.1807 | 0.1555 |
| | Semantic | 0.1561 | 0.2295 | 0.1913 | 0.0204 | 0.0539 | 0.0360 | 0.1016 | 0.1502 | 0.1247 |
| | **TopicSummRAG** | 0.2478 | 0.3303 | **0.2876** | 0.0383 | 0.0809 | **0.0588** | 0.1389 | 0.1871 | **0.1620** |

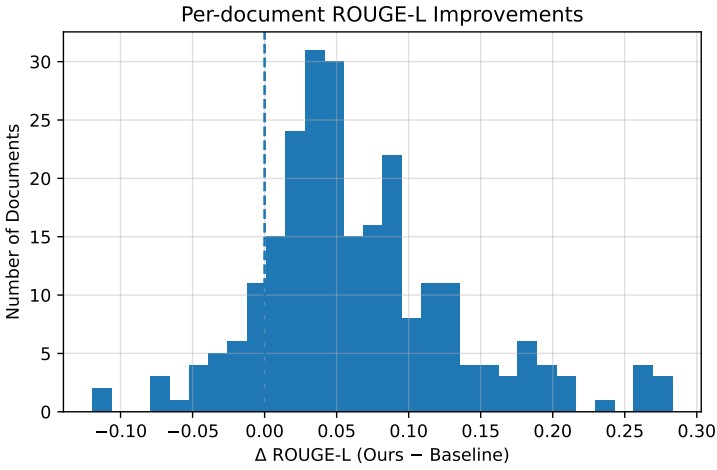

Figure 7: Distribution of per-document ROUGE-L improvements

Table 11: Extended Error Taxonomy for Query-Focused Summarization

| Code | Stage | Type | Description |
|---|---|---|---|
| *Retrieval-Level Errors* | | | |
| RE-01 | Retrieval | Miss | Key evidence required to answer the query is not retrieved. |
| RE-02 | Retrieval | Partial | Retrieved evidence covers only a subset of the information required by the query. |
| RE-03 | Retrieval | Noise | Retrieved segments contain substantial irrelevant or distracting content that obscures useful evidence. |
| RE-04 | Retrieval | Semantic Mismatch | Retrieved content is topically related but lacks the semantic evidence necessary to answer the query. |
| RE-05 | Retrieval | Granularity | Retrieved segments are too coarse or too fine-grained to support effective query-focused summarization. |
| *Summarization-Level Errors* | | | |
| SU-01 | Summarization | Omission | Relevant information present in the retrieved evidence is omitted from the generated summary. |
| SU-02 | Summarization | Query Misinterpretation | The summary reflects an incorrect understanding of the query intent (e.g., answering a different question). |
| SU-03 | Summarization | Evidence-Constrained Failure | The summary remains faithful to the retrieved evidence but fails to answer the query due to insufficient or misaligned information. |
| SU-04 | Summarization | Overgeneralization | The summary provides vague or generic statements instead of specific details required by the query. |
| SU-05 | Summarization | Over-Inclusion | The summary includes excessive background or tangential information, reducing focus on the query. |

### A.11.1 Error Taxonomy and Protocol

We developed an extended error taxonomy to disentangle failures at different stages of the RAG pipeline (Table 11). Retrieval-Level Errors (RE-01–05): Focus on the quality, noise, and granularity of the evidence provided to the LLM. Summarization-Level Errors (SU-01–05): Focus on the model's ability to use that evidence without omission, overgeneralization, or hallucination. Annotators were allowed to assign one or

more error codes per instance. In this setup, the absence of summarization-level errors (SU-01–SU-05) indicates that the summary is fully supported by the retrieved evidence.

### A.11.2 Retrieval Error Analysis

The retrieval-level performance (Tables 12 and 13) reveals how our structural design choices mitigate specific RE-series errors.

Table 12: Percentage distribution of QMSum retrieval errors

| Example | Sentence-level RAG (%) | TopicSummRAG (%) | Fixed (%) | Semantic (%) |
|---------|------------------------|------------------|-----------|--------------|
| RE-01 | 12.66 | 10.13 | 40.51 | 36.71 |
| RE-02 | 29.29 | 13.13 | 28.28 | 29.29 |
| RE-03 | 86.84 | 5.26 | 5.26 | 2.63 |
| RE-04 | 7.41 | 3.70 | 42.59 | 46.30 |
| RE-05 | 0.00 | 10.53 | 52.63 | 36.84 |

Table 13: Percentage distribution of ODSum retrieval errors

| Example | Sentence-level RAG (%) | TopicSummRAG (%) | Fixed (%) | Semantic (%) |
|---------|------------------------|------------------|-----------|--------------|
| RE-01 | 19.20 | 12.82 | 18.27 | 30.00 |
| RE-02 | 54.02 | 17.03 | 37.50 | 25.45 |
| RE-03 | 29.89 | 5.77 | 12.88 | 10.23 |
| RE-04 | 11.15 | 2.56 | 13.85 | 25.45 |
| RE-05 | 25.75 | 12.82 | 37.50 | 19.09 |

1. **Reducing Noise (RE-03):** Sentence-level RAG treats each sentence as a separate retrieval unit, while fixed-size chunking splits the document into predefined segments. Both approaches often include irrelevant content because they do not follow the natural structure of the document. As a result, Sentence-level RAG retrieves a large amount of noise (86.84% on QMSum). TopicSummRAG reduces this to 5.26% by retrieving coherent segments instead of arbitrary units, leading to cleaner evidence and less over-inclusion in the final summary.

2. **Reducing Semantic Mismatch (RE-04):** Semantic mismatch happens when retrieval relies on surface-level similarity rather than actual meaning. Semantic baselines still show high mismatch (46.30% on QMSum) because they lack a refinement step. TopicSummRAG lowers this to 3.70% by using segment-level summaries, which better capture the meaning of each segment and help avoid selecting text that appears similar but is not actually relevant to the query.

3. **Handling Granularity (RE-05):** Sentence-level RAG shows 0% error for RE-05, but this is mainly because it uses very small units and cannot adapt. In practice, it struggles to cover multi-part queries (e.g., 54.02% partial retrieval on ODSum). TopicSummRAG uses adaptive selection to adjust how many segments to retrieve based on relevance, improving coverage while keeping the context focused and leading to better summaries overall.

### A.11.3 Summarization Error Analysis

The downstream impact on the final summary is detailed in Table 14 and 15.

Table 14: Percentage distribution of QMSum summarization errors (lower is better)

| Error Code | Sentence-level RAG (%) | TopicSummRAG (%) | Fixed (%) | Semantic (%) |
|---|---|---|---|---|
| SU-01 | 45.45 | **20.23** | 52.33 | 43.43 |
| SU-02 | 12.50 | **7.24** | 16.28 | 15.15 |
| SU-03 | **0.00** | 2.30 | 2.33 | 3.03 |
| SU-04 | **4.55** | 9.20 | 29.07 | 38.38 |
| SU-05 | 37.50 | 21.03 | **0.00** | 0.00 |

**QMSum**

1. **Information Retention (SU-01):** Sentence-level RAG and Fixed miss a large portion of relevant information (over 45%), showing that retrieved evidence is not fully used in the final summary. TopicSummRAG reduces this by about 55% compared to Fixed, indicating that topic-based segmentation helps retain important information.

2. **Abstraction vs. Hallucination (SU-04):** Semantic and Fixed methods keep summaries short but tend to overgeneralize, losing important details. TopicSummRAG strikes a better balance, keeping summaries concise while still preserving key facts.

3. **Evidence Bottleneck (SU-03):** The near-zero values for SU-03 suggest that the models usually have enough evidence. The remaining issues come more from how the model uses the evidence during generation, rather than from missing retrieval.

Table 15: Percentage distribution of ODSUM summarization errors (lower is better)

| Error Code | Sentence-level RAG (%) | TopicSummRAG (%) | Fixed (%) | Semantic (%) |
|---|---|---|---|---|
| SU-01 | 38.98 | **21.67** | 39.29 | 36.43 |
| SU-02 | 10.17 | **3.33** | 4.46 | 4.65 |
| SU-03 | 12.71 | **8.33** | 14.29 | 13.18 |
| SU-04 | 27.97 | **17.00** | 25.00 | 36.43 |
| SU-05 | 10.17 | **7.67** | 16.96 | 9.30 |

**ODSUM**

1. **Narrative Continuity and Topic-Guided Retrieval:** TopicSummRAG is better at capturing the right parts of the text, as seen from the lower SU-01 (21.67%) compared to Sentence-level RAG (38.98%). The remaining errors mostly happen when the relevant information is subtle or spread across multiple parts of the document. The very low SU-02 (3.33%) shows that the method stays well aligned with the query and avoids drifting away from the intended topic.

2. **Syntactic and Semantic Challenges:** The values for SU-03 (8.33%) and SU-04 (17%) suggest that some issues still come from the generation stage. Even when retrieval is correct, the model sometimes produces slightly general or less precise statements, especially when the text contains multiple small subtopics. At the same time, TopicSummRAG keeps SU-05 low, meaning it avoids adding unnecessary background information, which is a common issue in story-based RAG.

### A.11.4 Case Studies: Qualitative Analysis

To illustrate these quantitative trends, we present a series of case studies. Case Study 1 and 2 illustrates the superior grounding of TopicSummRAG in complex discourse, while Case Study 3 identifies specific boundary

conditions where our topic-level aggregation may introduce trade-offs in precision. The content in `Green` indicates alignment with Gold Summary, whereas, in orange `Orange` indicates retrieval noise, semantic distortion, or overgeneralization.

Table 16: Legal divergence and procedural impact in Wales (Case Study 1).

| Field | Content |
| --- | --- |
| **Case Study 1** ||
| **Query** | Summarize the discussion about the divergence between the law in England and Wales. |
| **Gold summary** | Barry Hughes indicated that the Bill would **impact England** and their people, although it needed a period. Moreover, Barry thought there was no **cross-border issue** from the potential legislation, and the draft changing to **CPS guidance** would not be helpful for scrutinisers to **decide on the law** in Wales. |
| **Sentence-level RAG** | The discussion focused on the divergence between the law in England and Wales, specifically the `removal of the defence for reasonable punishment`. This change would mean that more cases would pass the evidential stage in Wales than in England, raising awareness issues for people from England who travel to Wales. The CPS recognizes the divergence and has a provision in their code to take account of potential changes. They plan to issue policy guidance and `charging standards` to reflect any changes. The guidance may need to be updated to reflect the change, particularly in relation to the public interest stage. The CPS is confident that their staff will be aware of the divergence, but may need to be brought up to speed on the specific changes. |
| **TopicSummRAG** | The discussion centered around the divergence between the law in England and Wales, specifically the removal of a defence that would make it more likely for the police to refer cases to the CPS in Wales. Barry Hughes stated that this would result in more cases passing the `evidential stage` in Wales than in England, raising awareness issues for people from England who travel to Wales. He also mentioned that the CPS already has a degree of divergence in laws, such as drink driving limits, and that they would need to update their guidance to reflect the changes. Hughes emphasized the importance of awareness among staff in Wales and England, and that they would need to be brought up to speed on the divergent laws. He also mentioned that there are `no other significant cross-border issues` arising from the potential legislation. |
| **Fixed** | The discussion revolves around the divergence of criminal law between England and Wales, specifically highlighting that this may be the `first time such a divergence has occurred`. The need to amend guidance to reflect this change was also discussed. |
| **Semantic** | There's been an obvious divergence between English and Welsh law, specifically in criminal law, which is the `first time this has happened`. The guidance needs to be amended to reflect this divergence. |

Case Study 1 (Table 16) focuses on a specific causal chain: the legislative removal of a defense leading to an increased volume of referrals at the evidential stage. While the standard RAG baseline identifies the start and end of this process, it misses the connection between the units required to explain the logic. Similarly, the Semantic and Fixed baselines collapse the reasoning into a single abstract statement, resulting in a loss of procedural detail. TopicSummRAG preserves this link because its Topic-Guided Segmentation ensures the "Evidential Stage" discussion remains anchored to the "Removal of Defense" context. By retrieving the full topical span rather than an arbitrary window, the model avoids omitting the logical bridge (SU-01) necessary for a faithful summary.

Table 17: Individual perspective retrieval (Barry Hughes) on CPS guidance (Case Study 2).

| Field | Content |
|---|---|
| **Case Study 2** | |
| **Query** | What did Barry Hughes think about draft changes to CPS guidance when talking about the divergence between the law in England and Wales? |
| **Gold summary** | Barry Hughes thought draft changes to CPS guidance would not necessarily be helpful to the scrutinisers, and simply wanted to take the **present interest factors**. Barry thought they needed to work for the principles of generality and a **degree of specificity**. |
| **Sentence-level RAG** | Barry Hughes mentioned that the removal of the defence would mean that there would be more cases that would pass the evidential stage in Wales than there would be in England. He also stated that the CPS would need to update the guidance to reflect the divergence between English and Welsh law, particularly in terms of awareness of criminal offending for people from England who travel to Wales. He agreed that the chances of the legislation passing are low, but it would provide a greater degree of protection for children and send a positive message. He suggested building an infrastructure for out-of-court disposals in Wales and raising awareness of the law through media publicity. |
| **TopicSummRAG** | Barry Hughes stated that the CPS guidance needs to be updated to reflect the divergence between English and Welsh law. He suggested that the existing guidance could be dealt with by making people aware of the defence not being available in Wales in the same way as in England. However, he believed that more specialist guidance would be needed for the public interest stage. Hughes also mentioned that staff in Wales are not yet fully aware of the divergence, but he expects them to be soon. He suggested that the CPS in England would also need to be aware of the changes, but not to the same extent as in Wales. |
| **Fixed** | Barry Hughes stated that his staff in Wales and England are not sufficiently aware of the divergence in the law as of the present day. |
| **Semantic** | Barry Hughes thinks the law should reflect contemporary society and be evolutionary in nature. He suggests updating the guidance, specifically at the evidential stage before the public interest stage, in response to draft changes to CPS guidance. |

Case Study 2 (Table 17) requires isolating Barry Hughes' specific perspective on CPS guidance updates. Here, standard RAG suffers from Over-inclusion (SU-05), pulling in irrelevant speculation about the likelihood of the bill passing. This occurs because window-based retrieval often grabs adjacent sentences that share high keyword similarity but lack functional relevance to the query. In contrast, TopicSummRAG uses Entropy-Driven Adaptive Retrieval to filter out these tangential segments. By selecting segments based on the dominance of the relevance distribution, the model ignores "media awareness" or speculative turns that do not directly address the query. This results in a summary strictly grounded in the evidence, satisfying the reviewer's requirement for a direct faithfulness audit.

Across both examples, the Semantic and Fixed baselines exhibit Semantic Distortion (SU-02/SU-04). Because these models rely on high-level embeddings, they often "reconstruct" summaries that sound fluent but introduce claims not present in the source—such as characterizing Hughes' view as "evolutionary in nature." TopicSummRAG mitigates this through its dual-Stage retrieval, the coarse stage identifies the correct topical discourse region, whereas, the fine-grained stage pinpoints the specific evidence sentences within that region. This ensures the generator is constrained by specific facts while maintaining enough topical context to avoid the overgeneralization seen in purely semantic models.

Table 18: Comparative focus on legal efficacy: Operational vs. Rights-based views (Case Study 3)

| Field | Content |
|---|---|
| **Query** | Summarize the discussion about the efficacy of the law. |
| **Gold summary** | Barry Hughes first stated that **children had fewer rights than adults** and therefore the law should be enforced to defend physical assault. As such social behavior was not available now, the law should change to reflect that. The discussion then turned to talk about the **legal framework** and its **prosecution**. |
| **Sentence-level RAG** | The discussion about the efficacy of the law revolves around its ability to `protect children`. Barry Hughes suggests that the law, as it presently stands, provides a greater `degree of protection`, even if there are low chances of prosecution. He argues that the law sends a message about what is acceptable and what is not, and that awareness is key to its success. Sian Gwenllian AM and Janet Finch-Saunders AM raise concerns about the potential barriers to implementing the proposed law, including the shortage of `registered intermediaries` who assist `child witnesses`. Barry Hughes acknowledges these concerns but emphasizes the importance of the law in setting a `framework for acceptable behavior`. |
| **TopicSummRAG** | The discussion about the efficacy of the law focused on the potential `unintended consequences` of introducing a new law in Wales. Barry Hughes from the Crown Prosecution Service (CPS) stated that the `numbers of prosecutions` for the proposed law would be small, and the CPS would not have targets for securing convictions. He also mentioned that the law could lead to an increase in `malicious reporting` against parents involved in private family law cases. However, he believed that the numbers would be tiny and that the CPS already has a well-developed approach to dealing with such cases. The `European Convention on Human Rights` was also discussed, with Hughes stating that implementing the legislation in Wales could make it more compliant with the Convention than England. He also emphasized the importance of awareness and education in understanding the new law, and that the criminal law provides a `general framework` within which to operate. |
| **Fixed** | The discussion revolves around the efficacy of the law, with concerns that it may not be effective if people are not prosecuted for breaching it. Individuals have argued that creating `legislation without enforcement is pointless`, suggesting a new law through the civil system could achieve `better culture change`. |
| **Semantic** | Individuals have argued that there is `no point creating legislation` if people are not prosecuted for breaching it. They questioned if changing the law would increase prosecution. |

Case Study 3 (Table 18) illustrates a divergence in Salience Attribution. While the Gold summary anchors the "efficacy of the law" in a rights-based framework—specifically the defense of children against physical assault—TopicSummRAG's entropy-based selection identifies a different discourse region. It prioritizes the operational and legal risks associated with the bill, such as the potential for malicious reporting and compliance with the European Convention on Human Rights (ECHR). This indicates a specific boundary condition: in highly heterogeneous discussions where a core concept (like "efficacy") is defined through multiple overlapping perspectives, the model may prioritize topical depth (technical/legal implications) over the thematic breadth (general social rights) captured by human annotators. Crucially, despite this shift in focus, TopicSummRAG remains strictly faithful to the source text it retrieved. Unlike the Fixed and Semantic baselines, which suffer from Overgeneralization (SU-04) by reducing the entire debate to a generic sentiment about enforcement, TopicSummRAG provides a high-fidelity technical summary of the retrieved segment.

This confirms that even in its worst cases, the failure mode is a matter of topical selection rather than a breakdown in factual grounding or a lack of contextual coherence.

Case Study 3 also highlights the trade-off noted in our limitation section. Because TopicSummRAG retrieves coherent topical spans rather than isolated sentences, the selected evidence includes broader surrounding context beyond the most localized aspect of the query. In this case, while the query targets a specific notion of "efficacy" (rights-based justification), the model retrieves a larger discourse segment centered on operational and legal considerations. This reflects a trade-off between maintaining discourse coherence and selecting minimally sufficient evidence for highly localized queries.

### A.12   Additional metrics and their correlation

To evaluate the causal link between our structural design and final performance, we analyze the pipeline as a tripartite flow: Topical Segmentation $\rightarrow$ Evidence Retrieval $\rightarrow$ Faithfulness in Summarization.

**Segmentation Evaluation**   The efficacy of TopicSummRAG is fundamentally based on the quality of the initial segment boundaries. To quantitatively evaluate the structural quality of our segmentation module, we compute Topic Purity using gold topic spans provided in the dataset. For each document, we compare predicted boundaries with ground-truth topic regions and construct predicted segments accordingly. A segment is considered pure if it overlaps with only one gold topic span. To account for minor boundary shifts, we apply a tolerance window of $\pm 2$ units when determining overlap. Topic Purity is then defined as the ratio of pure segments to total predicted segments. As shown in Table 19 (left), our framework achieves a Topic Purity of 0.703. This intermediate metric is critical as it indicates that over 70% of our predicted segments align strictly with a single ground-truth topic, preventing the contextual loss which is very common in fixed-window RAG. The high span-level overlap (Jaccard = 0.9063, Dice = 0.9438) ensures that the retrieved units are semantically unified. While boundary-level metrics (Pk, WindowDiff) show moderate sensitivity to minor shifts, the low Boundary Edit Distance (BED = 0.0726) confirms that the global topical structure remains intact.

Table 19: Performance of segmentation (left) and retrieval modules (right)

| Metric | Score |
|---|---|
| Pk ($\downarrow$) | 0.3579 |
| WindowDiff ($\downarrow$) | 0.3789 |
| Precision ($\uparrow$) | 0.1716 |
| Recall ($\uparrow$) | 0.5631 |
| F1 ($\uparrow$) | 0.2630 |
| BED ($\downarrow$) | 0.0726 |
| Jaccard ($\uparrow$) | 0.9063 |
| Dice ($\uparrow$) | 0.9438 |
| Topic Purity ($\uparrow$) | 0.703 |

| Metric | Score |
|---|---|
| Precision | 0.392 |
| Recall | 0.267 |
| Dice Coefficient | 0.317 |
| Jaccard Similarity | 0.189 |

**Retrieval Evaluation**   The retrieval performance in Table 19 is a direct consequence of the structural choices made during segmentation. To connect these aggregate metrics to our qualitative findings, we focus on two intermediate characteristics- query coverage defines the extent to which retrieved spans encompass the full ground-truth evidence set, while evidence redundancy defines the inclusion of topically adjacent but functionally non-essential discourse.

1. From Low Noise (RE-03) to High Query Coverage- The drastic reduction in RE-03 (Noise) from 86.84% in sentence-level RAG to 5.26% in TopicSummRAG is due to high topic purity (0.703). By

isolating the discourse segments, we eliminate the adjacent topic issue that typically forces generators to filter irrelevant data. This structural purity enables superior Query Coverage. In the Barry Hughes query (Table 20), TopicSummRAG identifies the exact discourse span ($[[17, 23]]$) required. Unlike window-based RAG, which often suffers from RE-02 (Partial Retrieval), our topic-aligned retrieval captures the complete discursive context of the testimony.. This ensures the generator LLM maintains the logical causal chain seen in the TopicSummRAG summary that directly contributes to the 0.95 Pearson correlation with ROUGE-L.

Table 20: Qualitative example showing high query coverage with TopicSummRAG.

| Component | GOLD | PREDICTED |
|---|---|---|
| **Example 1:** Query | What did Barry Hughes think about the implications for implementing the Bill in Wales when talking about divergence between England and Wales? | |
| Relevant Span IDs | $[[17, 23]]$ | $[21, 19, 18, 15, 17, 20, 16]$ |
| Summary | Barry Hughes concluded two implications: (1) the Bill would raise awareness for people from England traveling to Wales, and (2) others may reflect on this divergence similar to the drink driving limit in Scotland. | The removal of the defence in Wales would increase divergence from England, raise awareness issues, require guidance updates, and affect public interest considerations. The CPS already handles some divergence cases. |
| **Example 2:** Query | Summarize the discussion about microphone issues. | |
| Relevant Span IDs | $[[9, 18]]$ | $[9, 13, 11, 10, 14, 12]$ |
| Summary | The lapel microphones were considered too close and not representative. There was agreement that microphones capture breath and other non-voice sounds. | The discussion centered on lapel microphone placement, interference issues, mouth clicks, breath noise, and suggestions for alternative microphone configurations. |

2. From Granularity (RE-05) to Evidence Redundancy- The RE-05 (Granularity) rate of 10.53% highlights a specific trade-off of evidence redundancy. Because our units are self-contained segments, they occasionally include over-retrieved context that is topically grounded but functionally unnecessary for fine-grained queries. This trade-off is evident in the Auditor General case study (Table 21). Here, the model achieves high recall but introduces redundancy by pulling in adjacent details about staffing and budgets (RE-05). While factually grounded, this extra context causes contextual drift that shifts the summary focus from the core "parliamentary dissatisfaction" to secondary operational details. This demonstrates that while topical segments prevent hallucination, their inherent breadth can occasionally reduce the precision of the final summary.

Table 21: Qualitative example showing impact of evidence redundancy and contextual drift (RE-05)

| Component | GOLD | PREDICTED |
|---|---|---|
| **Example 1:** Query | What was said about the Auditor General? | |
| Relevant Span IDs | $[[49, 55]]$ | $[159, 163, 161, 150, 166, 51, 162, 167, 49, 52]$ |
| Summary | The Prime Minister was questioned about why sufficient funding was not provided to allow the Auditor General to conduct audits on government spending. The Prime Minister responded that the Auditor General's budget had been increased in the previous year. The response was considered unsatisfactory by the questioning party. | The Auditor General's office was described as lacking sufficient resources to perform its duties. Although funding was increased and additional staff were added, concerns remained regarding resource adequacy. The opposition continued to press the government to provide the necessary support. |

While the qualitative case studies illustrate how individual failures or successes relate to retrieval granularity and redundancy, they represent specific instances of a broader systemic pattern. To prove that these relationships are statistically consistent across the entire dataset, we conduct a formal correlation analysis in

the following section. This analysis shows how TopicSummRAG improves segmentation quality that leads to gain in final summarization performance.

### A.12.1 Correlation Analysis

**Topic-level evaluation protocol** To evaluate retrieval quality at the discourse level, we map turn-level relevance annotations to their corresponding topical segments. Since QMSum dataset provide ground-truth relevant spans as intervals (e.g., $[[1, 16]]$), we first expand each span into explicit turn IDs to ensure turn-level granularity. For each meeting, we construct a topic-to-turn mapping by expanding every topic's `relevant_text_span` and associating it with a unique topic identifier. Ground-truth and model retrieved turn IDs are then mapped to their corresponding topic IDs. A topic is considered successfully retrieved if at least one of its turns appears in the retrieved set. Evaluation is therefore performed over sets of topic IDs. We compute topic-level precision, recall, and F1 as:

$$Precision = \frac{|GT \cap Retrieved|}{|Retrieved|} \tag{1}$$

$$Recall = \frac{|GT \cap Retrieved|}{|GT|} \tag{2}$$

$$F1 = \frac{2 \cdot Precision \cdot Recall}{Precision + Recall} \tag{3}$$

where $GT$ and $Retrieved$ denote the sets of ground-truth and retrieved topic IDs, respectively. As shown in Table 22, the micro-averaged performance confirms that our system maintains high structural alignment at the topic level.

Table 22: Micro-Averaged Retrieval Performance

| Metric | Precision | Recall | F1 Score |
|--------|-----------|--------|----------|
| Micro  | 0.6138    | 0.7316 | 0.6675   |

**Correlation between Segmentation to Retrieval** To analyze the whether there is any relationship between topic coverage and fine-grained retrieval performance, we computed the correlation between Topic-level F1 and Retrieval F1@5. We observe a strong linear relationship (Pearson $r = 0.91$, $p < 0.001$) and a moderate-to-strong rank correlation (Spearman $\rho = 0.68$, $p < 0.001$). These results indicate that improvements in topic-level alignment are strongly associated with gain in retrieval performance. This strong association emerges naturally from the hierarchical architecture of TopicSummRAG. In our framework, topical segmentation determines the atomic retrieval units. When boundaries align with the document's latent discourse organization, segments exhibit high internal coherence and clear semantic separation. This structural purity concentrates query-relevant information and increases the margin between relevant and irrelevant segments in embedding space. The resulting similarity distribution is sharper, producing higher dominance scores during adaptive selection and limiting the propagation of noise. Conversely, misaligned boundaries shows relevance signals across mixed-topic segments, flattening similarity distributions and forcing the adaptive mechanism to retain broader, noisier candidate sets. In this sense, topical segmentation functions as a structural regularizer that governs semantic separability and retrieval breadth.

### A.12.2 Correlation between Retrieval to Summarization

We analyze whether improvements in retrieval quality lead to better summarisation performance by correlating F1@5 with standard generation metrics. For ROUGE-L vs F1@5, we observe a Spearman correlation

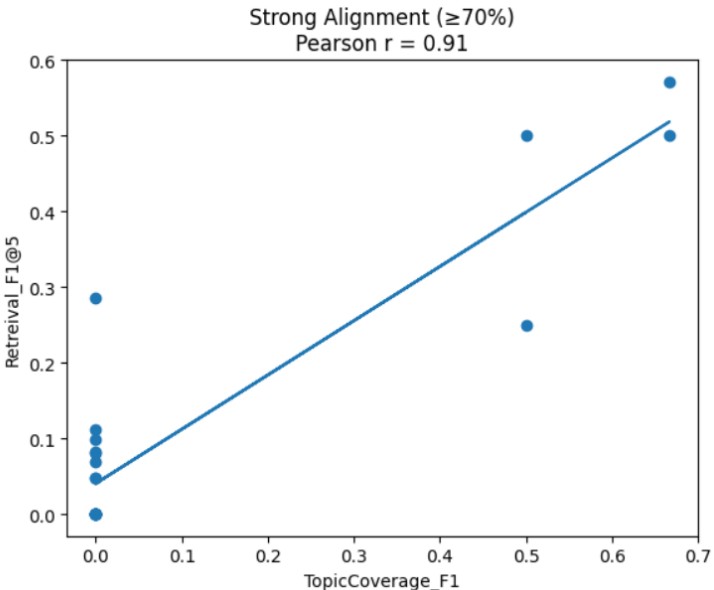

Figure 8: Correlation between Topic-level F1 and Retrieval F1@5.

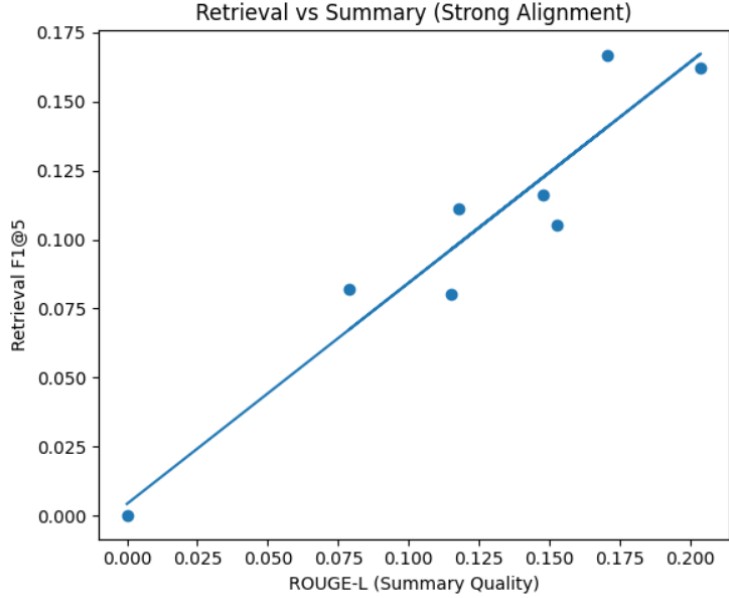

Figure 9: Correlation between Retrieval F1@5 and Summary ROUGE-L.

of $\rho = 0.8810$ ($p = 3.85e - 03$) and a Pearson correlation of $r = 0.9516$ ($p = 2.73e - 04$), indicating a near-linear relationship between retrieval effectiveness and summarisation quality. For BLEU vs F1@5, we obtain a Spearman correlation of $\rho = 0.8571$ ($p = 6.53e - 03$) and a Pearson correlation of $r = 0.7299$ ($p = 3.98e - 02$).

**Validation of the Tripartite Flow.** These findings empirically confirm the validity of the TopicSumm-RAG pipeline. By aligning retrieval units with coherent discourse spans, the system ensures that the evidence

provided to the generator is both topically grounded and contextually complete. The exceptionally high correlation with ROUGE-L indicates that gains in retrieval precision and recall translate directly into improved content coverage and sequence-level overlap in the final summary. This confirms that the bottleneck in long-document QFS is largely a retrieval-level challenge, when the structural alignment of evidence is preserved, the generated query focused summary is also better.

