# OpenReview forum: "TopicSummRAG: A Topic-Enhanced RAG model for Improving Query-Focused Summarization from Long Documents"
_TMLR — Rejected by TMLR_

### Review · Reviewer_uieT · 2026-02-09

**Summary Of Contributions:**

### Paper Summary

This paper proposes TopicSummRAG, which uses topic-aware segmentation to cut long dialogues or documents into chunks that better conform to the topic structure, for query-focused summarization in RAG scenarios. The method includes Contextually Coherent Topic Segmentation CCDS, segment-level summarization, and dominance-based adaptive retrieval to determine the retrieval granularity and the number of returned segments. The authors evaluated the method on datasets such as QMSum and ODSum using retrievers such as BM25, SBERT, and Situated Embeddings, and on multiple open-source LLM generators, reporting results for ROUGE, LENS, BLANC, G-EVAL, and some human evaluations. The overall conclusion is that TopicSummRAG produces more relevant and coherent summaries more consistently than fixed-size, sentence, and semantic chunking.

###  Strengths

1. Addressing the issue of incoherent evidence caused by naive chunking in RAG, the authors propose topic-aware indexing, a reasonable approach for long dialogue summarization.
2. The methodology is structurally complete. CCDS is used for boundary detection, followed by segment-level summarization and dominance-based adaptive retrieval.
3. The authors validated the model on multiple retrievers (BM25, SBERT, Situated), and multiple LLM generators.
4. The appendix includes sensitivity analysis and t-tests, along with error taxonomy and case studies.

### Weaknesses

1. The main claims regarding improvements in faciality and factual grounding currently lack direct and robust evidence. The paper heavily relies on LENS, BLANC, and G-EVAL to support improvements in fatality. However, these metrics are either learned metrics or LLM-as-a-judge, and cannot be equated to a manually verified factual consistency conclusion regarding source evidence. In particular, G-EVAL uses Gemini-2.0-Flash for zero-shot scoring; while a prompt is provided, it lacks control over the reliability and bias of the judge, such as consistency across multiple seeds, comparisons between different judge models, and analysis of systematic misjudgments when the judge does not cover the retrieved span adequately.
2. Human evaluation presents dimensions such as relevance, coherence, fatality, and fluidity, but the paper lacks sufficient information on the experimental design of human evaluation, resulting in insufficient persuasiveness of these results. For example, the number of reviewers, annotation guidelines, annotation consistency, whether blind sampling was used, sample size, sampling strategy, and statistical significance are not clearly explained in the available information, making it difficult to determine the reliability of the improvements in human fatality.
3. The causal chain between segmentation quality and downstream benefits is not clear enough. Table 3 provides abbreviations for boundary detection: P, R, F1, Pk, and WD. However, the main table's explanation of how segmentation errors are propagated to retrieval evidence quality and then to summary quality is currently more descriptive, lacking more direct evidence.
4. The baseline setting may be weak, affecting the credibility of the claims. The paper's comparison of chunking baselines mainly focuses on Fixed-size, Sentence, and Semantic. Semantic baselines also rely on off-the-shelf libraries; imprecise parameters and implementation details can lead to an underestimation of baseline performance. Furthermore, while the paper cites more recent chunking or segmentation-related work, it doesn't form strong comparative groups, resulting in insufficient evidence for the relative advantage of TopicSummRAG.
5. The error analysis includes taxonomy and case studies, but these are primarily qualitative presentations and cannot fully support the paper's claims regarding the scope of systematic improvements. For example, it's good that the authors distinguish between Retrieval-Level Errors and Summarisation-Level Errors in the appendix, but they lack statistical proportions of error types across the entire test set and comparisons of significant differences between different methods for each type of error. Therefore, it's difficult to use this to prove which type of error TopicSummRAG mainly reduces, whether it truly reduces hallucination, or whether it simply replaces it with over-inclusion and overgeneralization.

**Audience:**

Yes

**Audience Explanation:**

Some readers in the TMLR audience who work on Retrieval-Augmented Generation and long-document summarization may find the paper of interest, as it explores an alternative chunking strategy for handling long and topically structured inputs.

**Broader Impact Concerns:**

I do not see significant broader impact concerns beyond those common to Retrieval-Augmented Generation systems. Potential risks related to factual errors or misplaced trust in generated summaries are acknowledged implicitly, and the work does not introduce new ethical issues that would require an extensive Broader Impact Statement.

**Claims And Evidence:**

No

**Claims Explanation:**

See above.

**Requested Changes:**

1. Add a more direct faithfulness evaluation protocol, at least for manual fact-checking of QMSum or ODSum sampling. This should require annotators to judge whether the summary is supported solely based on retrieved evidence and report the error rate and consistency. This would ground the core claim of factual grounding in stronger evidence.
2. Make the design of the heuristic threshold for dominance-based adaptive retrieval more reproducible and supplement it with systematic comparisons of thresholds, such as the sensitivity of 0.6 vs. 0.4, or a comparison between learning-based gating and heuristic gating. Currently, a rule for k is provided, but evidence proving that this rule is necessary and optimal is lacking.

3. Add a more explicit analysis of the intermediate metrics link from segmentation to retrieval to summarization, such as topic purity, query coverage, or evidence redundancy metrics for the retrieved span, and demonstrate the correlation between these intermediate metrics and the final ROUGE LENS BLANC.

4. Provide stronger implementation transparency, such as the specific implementation of the Semantic chunking baseline, parameters, and whether it uses the same prompt and retrieval budget as TopicSummRAG, to avoid readers suspecting that the benefits come from the budget or that the prompt is inconsistent.

---

> ### Author Response · Authors · 2026-03-29
>
> We highly appreciate the detailed and valuable comments of the reviewer. We have truncated the requested changes (Q) and weaknesses (W) in order to fit within the character limit.
> > **Q1**: Add a more direct faithfulness evaluation protocol, at least for manual fact-checking ....
> >
> > **W1**: The main claims regarding improvements in faciality and factual grounding currently lack .......
> > **W2**: Human evaluation presents dimensions such as relevance, coherence, fatality, and fluidity .....
> > **W5**: The error analysis includes taxonomy and case studies, but these are primarily qualitative presentations   .......
>
> > **R1**: To address this, we have strengthened the evaluation in two ways.
> First, beyond automatic metrics, we conduct a manual audit on samples from QMSum and ODSum, where three annotators verify whether each generated summary is fully supported by the retrieved evidence (details in Appendix A.11). This evaluation is grounded in an explicit error taxonomy (RE-01–RE-05 and SU-01–SU-05), allowing us to quantify different types of retrieval- and summarization-level errors across the dataset. We report the distribution of these error types across multiple baselines (Sentence-level RAG, Fixed, Semantic, and TopicSummRAG), as presented in Appendix (Sections A.11.2 and A.11.3). From the retrieval side, we observe a clear reduction in noisy and incomplete evidence. For example, on QMSum, Sentence-level RAG retrieves a large amount of irrelevant content (86.84% noise), while TopicSummRAG reduces this to 5.26%. On ODSum, standard RAG often fails to cover all aspects of a query (54.02% partial retrieval), whereas our method reduces this to 17.03%, leading to better coverage. These improvements directly translate to the summarization stage, on QMSum, the omission of relevant information (SU-01) drops from 45.45% to 20.23%, and we observe fewer cases of query misalignment (e.g., 3.33% vs. 10.17% on ODSum). Overall, TopicSummRAG produces summaries that remain more tightly grounded in the retrieved evidence, while avoiding the high over-generalization observed in semantic baselines.
> The analysis also highlights trade-offs. In particular, TopicSummRAG shows a moderate increase in over-inclusion (SU-05) in QMSum, which reflects the use of broader topical segments. However, this trade-off leads to improved coverage and fewer omissions overall, suggesting a better balance between completeness and precision.
> Second, we have expanded the description of the human evaluation protocol to improve transparency and reproducibility, including details on sampling, annotation setup, and agreement measures. Regarding G-EVAL, we now explicitly position it as a supporting metric rather than primary evidence. We acknowledge its limitations as an LLM-based judge and do not rely on it alone to claim factual improvements. Instead, our conclusions are supported primarily by the manual evaluation and error analysis, with automatic metrics serving as complementary signals.
> We also include qualitative case studies (Section A.11.4) to illustrate these patterns. Taken together, the quantitative analysis and examples provide direct evidence that the improvements stem from better evidence selection and alignment, rather than changes in generation behaviour alone.
> ---
>
> > **Q2**: Make the design of the heuristic threshold for dominance-based adaptive retrieval more reproducible .....
>
> >**R2**: In the revised manuscript, we include a sensitivity analysis of the thresholds $(t_1, t_2)$ (Figure 7, bottom-right), where we evaluate performance over a grid spanning $t_1 \in [0.3, 0.7]$ and $t_2 \in [0.2, 0.6]$. Rather than relying on a few fixed configurations, this allows us to examine how performance behaves across the threshold space more broadly. We observe that performance varies smoothly across this range and does not exhibit a sharp optimum. Instead, there is a broad region where many threshold combinations yield similar results. This suggests that the proposed rule is not sensitive to precise tuning and is not tied to a specific choice such as $(0.6, 0.4)$. We further support this with our precision–recall analysis, which shows that adaptive retrieval consistently outperforms fixed choices of $k$. This highlights the importance of dynamically selecting $k$ based on the relevance distribution, rather than relying on a fixed retrieval budget. Overall, these results indicate that the dominance-based selection strategy is robust and effective in balancing precision and recall, without requiring careful tuning of thresholds.
> ---
> Kindly note: All the changes in the updated manuscript are highlighted in blue for the ease of reviewing.

---

> ### Author Response · Authors · 2026-03-29
>
> > **Q3**: Add a more explicit analysis of the intermediate metrics link from segmentation to retrieval to summarizatio ....
> >
> > **W3**: The causal chain between segmentation quality and downstream benefits .......
>
> > **R3**: To address this, we have added a new section, "Additional Metrics and Their Correlation" (Appendix A.12), which provides a tripartite analysis of the information flow:
> Segmentation Quality: We introduced Topic Purity ($0.703$) to measure how well our predicted segments align with ground-truth discourse units. We show that high purity prevents contextual loss within the segments which is a primary source of noise in fixed-window RAG.
> Retrieval Quality: We quantified the link between segmentation and retrieval through Query Coverage and Evidence Redundancy. We demonstrate that our reduction in retrieval noise (RE-03) from $86.84\%$ to $5.26\%$ is a direct result of this topic purity.
> Statistical Correlation: We computed the Pearson and Spearman correlations across the entire pipeline. We found a near-linear relationship between segmentation and retrieval (Pearson $r=0.91$), whereas, Retrieval F1 and Summarization quality (Pearson $r=0.95$).
> Qualitative analysis: In addition to these quantitative results, we provide qualitative examples showing how segmentation errors lead to either incomplete retrieval (missing evidence) or redundant retrieval (extra context), which in turn affects summary quality.
>
> ---
>
> > **Q4**: Provide stronger implementation transparency, such as the specific implementation .....
> >
> > **W4**: The baseline setting may be weak, affecting the credibility of the claims. ....
>
> >**R4**: We have revised Section 4.2 to provide more explicit details about all baseline setups. In particular, for the Semantic chunking baseline, we now describe the full implementation: we use the *all-mpnet-base-v2* embedding model, a 3-sentence sliding window, and breakpoint selection based on the 95th percentile of cosine distance. These settings are adapted from prior work and standard implementations, and were chosen to provide a competitive baseline rather than a weak one.
> To ensure fairness across all methods, we strictly control the experimental setup. All RAG variants use the same retrieval model, generation backbone (LLaMA, Gemma, Qwen), prompt template. This matches the settings used by TopicSummRAG, ensuring that improvements are not due to differences in prompt design or available context, but come from the retrieval strategy itself.
> Beyond the basic chunking baselines (Fixed-size, Sentence, Semantic), we have also expanded our comparisons to include stronger segmentation approaches (e.g., MiniSeg, DialogLM, UPS) as well as query-focused summarization baselines spanning zero-shot, supervised, and RAG-based methods. We further include comparisons with recent work such as *Document Segmentation Matters for Retrieval-Augmented Generation*, which provides a more relevant point of comparison for segmentation in RAG settings.
> Overall, these additions make the evaluation more transparent and comprehensive, and better position TopicSummRAG against both standard baselines and recent methods.

---

> ### Author Response · Authors · 2026-04-08
>
> We would like to thank the reviewer again for the valuable feedback. As the discussion period draws to a close tomorrow, we wanted to reach out to ensure that our previous responses have adequately addressed your concerns. We remain fully available to clarify any remaining points or answer additional questions you might have.

---

### Review · Reviewer_4pE5 · 2026-02-28

**Summary Of Contributions:**

This paper presents a RAG-based approach for Query-Focused Summarization from long documents. While prior work typically adopts fixed-size sliding windows or sentence-level units as chunks (i.e., retrieval units in RAG), this study proposes using topic-based units instead.

In the proposed method, positive and negative pairs are automatically constructed from domain-specific texts and used as supervision signals to fine-tune a text encoder, unsup-simcse-roberta-base, with a triplet loss objective. The text is first segmented into units (e.g., sentences, paragraphs, dialogue turns). The cosine similarity between embeddings of adjacent units is then computed, and topic boundaries are determined using Otsu thresholding and a gradient-based threshold. This process yields topic-aware chunks. After semantic coherent pruning, the final topic segments are obtained. For each segment, a summary is generated using Qwen3-8B. These summaries are encoded with SBERT and used as retrieval targets in the RAG framework.

For Query-Focused Summarization, the query is encoded with SBERT, and the top-L most similar segments are retrieved. The similarity scores are converted into a probability distribution using a temperature parameter, and the shape of this distribution is used to determine how many segments should be selected as sources for summarization. From the selected segments, sentences that are most similar to the query, according to an SBERT-based dense retriever, are further extracted and provided to the LLM to generate the final summary.

The evaluation consists of two tasks: topic segmentation and query-focused summarization. For topic segmentation, experiments are conducted on QMSum and TIAGE, where the proposed method is reported to outperform existing approaches such as SumSeg and TopSeg. For query-focused summarization, ODSum-Story, QMSum, and ODSum-Meeting are used as evaluation datasets, and Qwen3-8B, Llama 3.1 8B, and Gemma3-12B are employed as summarization models (LLMs). When varying the retrieval unit in RAG, the proposed method achieves the best performance. In addition, ablation studies are conducted to analyze the impact of elements in the proposed method on topic segmentation and retrieval accuracy within the RAG framework.

**Additional Comments:**

Section 1 contains very few citations. While I am familiar with this area and therefore understand the background, the audience of this journal is primarily in machine learning. The discussion should be grounded more clearly in prior work, with appropriate citations used to contextualize the problem setting, motivate the proposed approach, and clarify how it relates to existing research.

Without more detailed explanations of Otsu thresholding and gradient-based thresholding, I am not confident about the method. Please provide additional implementation details.

In the Semantic Coherence Pruning step, I could not clearly understand the principle behind quantifying coherence using the L2 norm of embeddings.

Figure 3 does not explain what the methods labeled "Fixed-size," "Sentence," and "Semantic" actually correspond to. While "Fixed-size" and "Sentence" are relatively self-explanatory, it is unclear what "Semantic" specifically refers to, and how it differs from the proposed method. The paper should clarify the exact segmentation strategy used for "Semantic" and how it is distinct from the topic-aware segmentation introduced in this work.

In Table 3, for "TopicSumm + E → RAG," the reported scores are P = 0.39, R = 0.26, and F1 = 0.27. Is this F1 score computed correctly? Given these precision and recall values, the harmonic mean would be substantially higher than 0.27. The authors should verify the calculation and clarify how F1 is computed.

**Audience:**

No

**Audience Explanation:**

The proposed method consists of (1) topical segmentation, (2) topic-aware coarse retrieval using segment-level summaries, (3) entropy-based adaptive selection, (4) sentence-level fine-grained retrieval within selected regions, and (5) final answer generation using a RAG model. However, I could not understand from the paper why this particular pipeline is necessary for query-focused summarization (merely explaining the need for topic boundary detection or retrieval is insufficient). In Section 3, the justification for adopting this design is weak. It might be acceptable if experiments demonstrated that this composition is the strongest option, but the ablation study in Section 4.2 is not about query-focused summarization; it focuses on topic segmentation and retrieval accuracy. My impression is that a standard RAG-based summarizer, where one adjusts the segmentation granularity (e.g., sentence, paragraph, dialogue turn) based on the dataset and tunes hyperparameters such as the number of retrieved items, could already be a strong baseline, yet there is no experimental comparison against it.

I also do not find the topic segmentation component particularly novel. The idea of detecting boundaries based on text similarity has existed for over 20 years (e.g., the lecture slides in [1]). Replacing word-overlap measures with sentence embeddings is a straightforward extension (and there are likely existing methods along these lines). While the paper also uses specific thresholding strategies and semantic-coherent pruning, it would be difficult to argue that these choices constitute substantial novelty, and the authors do not discuss novelty in Section 3.1.

[1] https://ocw.mit.edu/courses/6-892-computational-models-of-discourse-spring-2004/434c44c4c6c8f51d1a08d5b483894c1d_lec02.pdf

**Claims And Evidence:**

No

**Claims Explanation:**

Although the paper is not difficult to read, some parts are unclear due to insufficient explanation of the methodological details and the underlying motivation. That said, most parts of the proposed method can be understood from the description. However, the necessity and novelty of the approach are not sufficiently justified, and in particular, the empirical validation for the query-focused summarization task appears inadequate.

TopicSummRAG explicitly performs fine-tuning on QMSum. While the paper does not clearly specify where the positive and negative examples in P4 are drawn from, it seems likely that gold topic boundaries are used to construct these examples, effectively tuning the model for the topic segmentation task itself. Alternatively, unit adjacency may serve as a supervision signal to adapt the embeddings to the target domain. In contrast, methods such as SumSeg, MiniSeg, and BiLSTM_RoBERTa do not appear to perform fine-tuning on the target domain (though this is not entirely clear from their papers). Therefore, it is difficult to determine whether the improvements shown in Table 2 truly demonstrate the superiority of the proposed method, or simply reflect the effect of domain-specific tuning. The comparison does not allow us to disentangle these factors.

For evaluation on the TIAGE dataset, Utterance-Pair Modeling + Pretraining (UPS) (Yang et al., NAACL 2025) should also be included as a baseline.

Regarding the query-focused summarization task, the paper reports experimental results under different configurations of topic segmentation and retrieval methods. However, since there is no direct comparison with prior work in this task, the results do not provide sufficient evidence to support the claimed superiority of the proposed approach.

Shihao Yang, Ziyi Zhang, Yue Jiang, Chunsheng Qin, and Shuhua Liu. 2025. A Unified Supervised and Unsupervised Dialogue Topic Segmentation Framework Based on Utterance Pair Modeling. In Proceedings of the 2025 Conference of the Nations of the Americas Chapter of the Association for Computational Linguistics: Human Language Technologies (Volume 1: Long Papers), pages 4898–4908, Albuquerque, New Mexico. Association for Computational Linguistics.

**Requested Changes:**

+ Compare the proposed method with previous studies and simple baselines on query-focused summarization.
+ Construct a justification for the proposed method from the perspective of research on query-focused summarization.
+ Explain the novelty and originality of the work in Section 3.
+ Provide detailed descriptions of the baseline methods used in the experiments, so that readers can clearly understand whether the experimental results appropriately validate the effectiveness of the proposed ideas.

---

> ### Author Response · Authors · 2026-03-29
>
> We highly appreciate the detailed and valuable comments of the reviewer. We have truncated the requested changes (Q) and weaknesses (W) in order to fit within the character limit.
> > **Q1**: Compare the proposed method with previous studies and simple baselines on query-focused summarization.
> >
> > **Q4**: Provide detailed descriptions of the baseline methods used in the experiments...
> >
> > **W4**: Regarding the query-focused summarization task, the paper reports experimental results under different configurations…
>
>
> >**R1**: In the revised manuscript, we have expanded the experimental section to include a broader and more structured set of baselines (Section 4.2.2). For QFS, we now cover zero-shot LLMs (e.g., LLaMA-3.1-8B, Qwen-3-8B, Gemma-3), structured approaches such as chapterization-based summarization, supervised models (e.g., BART-Large, DialogLM), and RAG-based methods. This allows us to position our method across different levels of supervision and architectural design.
> We also include simple yet strong RAG baselines (Fixed-size, Sentence, and Semantic chunking) under a controlled setting where the retrieval model, generation backbone, prompt, and token budget are kept fixed. This ensures that performance differences are primarily due to the segmentation strategy rather than other factors.
> Additionally, we have revised the baselines section of Topic Segmentation to categorize baselines based on their learning paradigm (supervised vs. unsupervised)  (Section 4.1.2). We also clarify key differences in supervision, input handling, and architectural design across methods.
> These changes make the experimental setup more transparent and ensure that comparisons against both prior work and simple baselines are fair and well-defined. The results show that TopicSummRAG consistently outperforms standard zero-shot and simple RAG baselines, while achieving performance close to supervised methods on QMSUM, and generalizes well across datasets such as ODSum-Story and ODSum-Meeting without task-specific fine-tuning.
>
> ---
>
> > **Q3**: Explain the novelty and originality of the work in Section 3.
> >
> > **W7**: I also do not find the topic segmentation component particularly novel. ....
> >
>
> >**R3**: Our goal is not to claim novelty in the basic boundary detection signal. Instead, the main contribution of our work lies in how segmentation is formulated and used within query-focused summarization (QFS).
> First, we treat segmentation as a retrieval-driven operation, rather than a standalone task. In prior work, segmentation is typically evaluated in isolation (e.g., boundary accuracy). In contrast, our objective is to produce segments that are useful for retrieving coherent, query-relevant evidence, which leads to a different design focus.
> Second, we do not treat segmentation as a preprocessing step. It is integrated into a multi-stage retrieval pipeline, where segment-level summaries, adaptive selection, and fine-grained retrieval work together. Our ablation study shows that segmentation alone is insufficient—its interaction with these stages is what improves performance.
> Third, we provide empirical evidence for this design. We show that better segmentation is strongly associated with improved retrieval, and that retrieval quality is in turn linked to better summaries. We also observe consistent reductions in summarization-level errors (Appendix A.11.2), particularly for omission and query misalignment.
> Finally, regarding terminology, we use the term “topic” to refer to locally cohesive discourse segments, rather than global or labeled topics. In practice, our method performs segmentation into coherent units that are internally consistent and distinct from their neighbors, which is more aligned with discourse segmentation than classical topic modeling.
> We have revised Section 3.1 to make this positioning clearer and to explicitly state that our contribution is in adapting and integrating segmentation for QFS, rather than proposing a fundamentally new segmentation algorithm. To make this intuition more concrete, we also include a brief illustrative example in the Introduction demonstrating how segmentation improves query-focused evidence aggregation.
>
> Kindly note: All the changes in the updated manuscript are highlighted in blue for the ease of reviewing.

---

> ### Author Response · Authors · 2026-03-29
>
> > **Q2**: Construct a justification for the proposed method from the perspective of research on query-focused summarization....
> >
> > **W6**: The proposed method consists of (1) topical segmentation, (2) topic-aware coarse retrieval using segment-level summaries, ...
>
>
> > **R2**: A key challenge in QFS is identifying and combining query-relevant evidence from long, multi-topic documents, where relevant information is often scattered across different parts of the text. Prior approaches either rely on global encoding (which struggles with long contexts) or retrieval over fixed or heuristic chunks. In practice, these chunking strategies tend to either break coherent discourse or mix unrelated content, leading to fragmented or noisy evidence.
> Our approach is based on the idea that retrieval units should align with coherent regions of discourse, rather than arbitrary chunks. We use the term “topic” in a loose sense here—it does not refer to a labeled topic or a globally consistent theme, but rather to a locally cohesive segment of text that is internally consistent and meaningfully different from what comes next. In that sense, this is closer to discourse segmentation than traditional topic modeling. This distinction matters in QFS settings. For instance, a discussion about a specific issue (e.g., UI problems in a meeting) may appear in multiple, non-contiguous parts of the document. Our segmentation does not try to merge these into a single global topic; instead, it treats them as separate but coherent segments. At retrieval time, segment-level summaries allow the system to pick up all relevant pieces, even when they are spread across the document.
> More broadly, our design is motivated by a limitation we repeatedly observed with standard RAG pipelines: the trade-off between fragmentation and noise. Smaller chunks tend to improve precision but often miss relevant context that is distributed across the document. Larger chunks improve coverage but bring in irrelevant content. In practice, simply tuning chunk size or retrieval depth does not fully resolve this issue. Our pipeline separates these steps—first retrieving broader, coherent segments for coverage, and then filtering them to improve precision—which is difficult to achieve in a single-stage setup. We support this design with additional analysis in the Appendix Section A.12. The induced segments align well with discourse structure (Topic Purity = 0.703, Jaccard = 0.9063, Dice = 0.9438). We also observe that better segmentation is strongly associated with improved retrieval (Pearson $r = 0.91$), and that retrieval quality is closely tied to summarization performance ($r = 0.95$). The ablation study Section 4.2.5 further shows that segmentation alone leads to high recall but also introduces noise, whereas the full pipeline produces a better balance of precision and recall. Finally, the error analysis shows consistent reductions in retrieval noise (RE-03), partial retrieval (RE-02), and summarization-level omissions (SU-01), indicating that improvements in retrieval quality translate into more complete and better-grounded summaries. Overall, our contribution is not to introduce a new notion of “topic,” but to use segmentation as a practical way of isolating coherent discourse units, and to integrate it within a multi-stage retrieval pipeline that improves evidence selection and grounding in QFS. We have revised Section 3 to make this motivation clearer.
>
> ---
>
> >**W2**: TopicSummRAG explicitly performs fine-tuning on QMSum. While the paper does not clearly specify...
> >
> >**W3**: For evaluation on the TIAGE dataset, Utterance-Pair Modeling + Pretraining (UPS)...
>
> >**R5**: We have updated Section 4.1.2 to include UPS (Yang et al., 2025) as a supervised baseline on the TIAGE dataset.
> To clarify the concern regarding domain-specific tuning, our approach is fully unsupervised and does not use gold topic boundaries from QMSum or TIAGE during training. The contrastive embedding model is trained once and applied across datasets without any dataset-specific fine-tuning. In contrast, UPS is explicitly trained on the TIAGE distribution via utterance-pair supervision, which allows it to better align with dataset-specific boundary annotations.
> To make this distinction clear, we have revised the experimental section to explicitly separate supervised / fine-tuned methods from unsupervised / untuned methods (Section 4.1.2). Under this setting, UPS achieves the strongest performance on TIAGE, as expected for a supervised model. However, our method remains competitive among unsupervised approaches and demonstrates consistent performance across datasets without requiring labeled data.
> We also note that our ablation study (Table 5 (left))  shows that removing the contrastive embedding component leads to a noticeable drop in segmentation performance, suggesting that the gains stem from the proposed representation and segmentation strategy rather than dataset-specific adaptation.

---

> ### Author Response · Authors · 2026-03-29
>
> >**Q5**: Without more detailed explanations of Otsu thresholding and gradient-based...
>
> >**R4**: In our approach, we first compute cosine-depth scores and identify local maxima as candidate boundary points. The depth values are treated as a 1D distribution over the document, and we apply Otsu thresholding to retain only those peaks that are globally salient. Specifically, Otsu’s method determines a threshold that separates lower depth values (corresponding to within-topic continuity) from higher depth values (corresponding to potential topic transitions), allowing us to avoid manual threshold selection. In addition, to capture locally abrupt changes, we compute the first-order difference of the depth curve and retain positions with relatively high gradient magnitude. This step helps distinguish sharp semantic shifts from gradual topical drift. Since these two signals may not perfectly align due to windowing effects, we reconcile them within a small tolerance window (±1–2 positions) and retain only those candidates that satisfy both criteria. This ensures that selected boundaries are both globally prominent in the depth distribution and locally indicative of abrupt semantic change.
> We have updated the manuscript (Section 3.1) to include these details more clearly.
>
> ---
>
> > **Q6**: In the Semantic Coherence Pruning step, I could not clearly understand the principle behind quantifying coherence using the L2 norm of embeddings.
> >
>
> >**R6**: In our approach, we use the $\ell_2$ norm of embeddings as a simple way to check how consistent the representations are within a segment. Since the embeddings are trained with a contrastive objective, units from the same segment tend to have similar representation patterns. The coherence score looks at how stable these norms are within a segment. If the segment is consistent, the norms are fairly similar; if there is a topic shift, the values tend to vary more. We capture this by comparing each unit’s norm with the segment average, which gives a simple measure of consistency. This is not meant to be a precise semantic measure, but rather a lightweight check to see whether splitting a segment improves its internal consistency.
> We have revised the manuscript (Section 3.1) to explain this more clearly.
> ---
>
> >**Q7**: In Table 3, for "TopicSumm + E → RAG," the reported scores are ... Is this F1 score computed correctly? ...
>
> >**R7**: The F1 score in Table 5 is not computed directly from the reported aggregate precision and recall. Instead, we use a macro-averaging setup, where F1 is computed separately for each document and then averaged across all documents. This is why the reported F1 (0.27) appears lower than expected if one combines P and R directly.
> We have clarified this in the Table 5 caption in the revised manuscript to avoid confusion.
>
> ---
>
> >**Q8**: Figure 3 does not explain what the methods labeled "Fixed-size," "Sentence," and "Semantic" actually correspond to. ...
>
> >**R8**: For Fixed-size, we split the document into chunks of a predefined token length, without considering linguistic or semantic boundaries. For Sentence, each sentence is treated as an independent retrieval unit. For Semantic chunking, we group sentences based on embedding similarity. In our implementation, we use the all-mpnet-base-v2 model, compute cosine distance over a sliding window of 3 sentences, and introduce breakpoints when the distance crosses a 95th percentile threshold, resulting in variable-length segments. The key difference is that semantic chunking focuses on local similarity, while our method aims to capture more global, discourse-level structure. In TopicSummRAG, segments are formed based on broader thematic consistency, which helps preserve longer-range context that local similarity-based methods often miss.

---

> ### Author Response · Authors · 2026-04-08
>
> We would like to thank the reviewer again for the valuable feedback. As the discussion period draws to a close tomorrow, we wanted to reach out to ensure that our previous responses have adequately addressed your concerns. We remain fully available to clarify any remaining points or answer additional questions you might have.

---

### Review · Reviewer_KR2X · 2026-03-26

**Summary Of Contributions:**

This paper proposes TopicSummRAG, a novel framework designed to enhance Retrieval-Augmented Generation for query-focused summarization over long, multi-topic documents.

The paper introduces a highly intuitive and principled approach to document chunking by aligning retrieval units with latent topical structures rather than arbitrary fixed sizes or sentence-level heuristics.

**Audience:**

Yes

**Audience Explanation:**

Yes, researchers and practitioners within the TMLR audience would likely find the findings of this paper highly relevant, particularly those focused on NLP and LLMs.

**Claims And Evidence:**

Yes

**Claims Explanation:**

The approach is evaluated extensively on benchmark datasets, including QMSum and TIAGE for the segmentation component , as well as ODSum-Story, ODSum-Meeting, and QMSum for the summarization task.

The empirical results demonstrate substantial improvements over strong baselines across multiple retrievers and large language models.

**Requested Changes:**

The authors acknowledge the additional computational overhead introduced by generating segment-level summaries, but the paper currently lacks a detailed quantitative comparison of preprocessing costs or end-to-end latency. Adding a brief quantitative analysis on latency would strengthen the practical evaluation of the system.

The limitation section notes that highly localized queries may occasionally include limited surrounding context, highlighting a trade-off between discourse coherence and minimal evidence selection. Providing a more prominent, concrete example of this trade-off in the main text could help readers better understand the system's boundary conditions.

---

> ### Author Response · Authors · 2026-04-07
>
> We highly appreciate the detailed and valuable comments of the reviewer.
> > **Q1**: The authors acknowledge the additional computational overhead introduced by generating segment-level summaries, but the paper currently lacks a detailed quantitative comparison of preprocessing costs or end-to-end latency. Adding a brief quantitative analysis on latency would strengthen the practical evaluation of the system.
> >
>
> > **R1**: In the current work, we focus primarily on improving retrieval quality and generation performance. Hence, the comparative analysis are focused on the same. In order to provide a sense of execution time, we now report the total time taken by our method along with an independent component-wise latency analysis (Section 4.2.4). On average, a single query requires 30.05 seconds end-to-end, with final answer generation accounting for 28.11 seconds, topic-aware coarse retrieval with entropy-based adaptive selection accounting for 1.93 seconds, and sentence-level fine-grained retrieval within matched regions contributing negligible latency (<0.01 seconds). Accordingly, we modified the limitation section.
>
> ---
>
> > **Q2**: The limitation section notes that highly localized queries may occasionally include limited surrounding context, highlighting a trade-off between discourse coherence and minimal evidence selection. Providing a more prominent, concrete example of this trade-off in the main text could help readers better understand the system's boundary conditions.
> >
>
> > **R2**: In the revised manuscript, we have added an explicit discussion in Section A.11.4 (Case Study 3) illustrating this boundary condition. Specifically, we show that for queries targeting a localized notion of ``efficacy,” TopicSummRAG retrieves a broader, topically coherent segment that includes surrounding operational and legal context. This demonstrates the trade-off between maintaining discourse coherence and selecting minimally sufficient evidence for highly localized queries. We believe this addition makes the system’s boundary conditions more transparent and better grounded in empirical examples.
>
>
> Kindly note: All the changes in the updated manuscript are highlighted in blue for the ease of reviewing.

---

> ### Author Response · Authors · 2026-04-08
>
> We would like to thank the reviewer again for the valuable feedback. As the discussion period draws to a close tomorrow, we wanted to reach out to ensure that our previous responses have adequately addressed your concerns. We remain fully available to clarify any remaining points or answer additional questions you might have.

---

### Decision · Action_Editor_iSCf · 2026-04-30

**Recommendation:** Reject

**Audience:**

Yes

**Audience Explanation:**

Researchers and practitioners within the TMLR audience who are focused on NLP and LLMs would likely find the findings highly relevant.

**Claims And Evidence:**

No

**Claims Explanation:**

The reviewer felt the empirical validation for the query-focused summarization task was inadequate. They noted a lack of direct comparison with prior work and expressed concerns that the improvements might simply reflect the effect of domain-specific tuning rather than the superiority of the proposed method. The reviewer argued that the claims regarding improvements in factual grounding lacked robust evidence, as the paper heavily relied on automated LLM-as-a-judge metrics without sufficient manual fact-checking. They also pointed out that the human evaluation lacked experimental design details, the baseline settings may have been too weak, and the causal link between segmentation quality and downstream benefits was not clearly established.

---

> ### Author Response · Authors · 2026-05-05
>
> We thank the Action Editor for summarizing the reviewers' concerns. However, we would like to point out that the meta-review is based on the original reviews provided initially. We have made substantial revisions in our manuscript to address all of these concern including new experiments to support our claims. For example, regarding the concern about lack of direct comparison with prior work and potentially weak baselines, we have significantly expanded the experimental section (Section 4.2.2). The revised version now includes comparisons with zero-shot LLMs, supervised models (e.g., BART-Large, DialogLM), and multiple RAG-based baselines, along with standard chunking approaches (Fixed-size, Sentence, Semantic). Importantly, all methods are evaluated under a controlled setting (same retriever, generator, prompt, and token budget), ensuring that improvements are attributable to the proposed method rather than differences in configuration. This directly addresses Reviewer 1’s concerns (Q1, W4).
>
> To clarify the concern that improvements may come from domain-specific tuning, we have explicitly stated in Section 4.1.2 (R5) that our method is fully unsupervised and does not rely on dataset-specific supervision. We also include stronger supervised baselines (e.g., UPS) to clearly separate the effect of supervision from the proposed approach, addressing Reviewer 1’s concern (W2).
>
> For the issue of factual grounding and reliance on LLM-based metrics, we have strengthened the evaluation by adding a manual fact-checking study with three annotators (Appendix A.11). In this analysis, we explicitly evaluate both retrieval-level and summarization-level errors using a structured error taxonomy, and compare these error distributions across different RAG baselines (Fixed-size, Sentence, Semantic, and TopicSummRAG). We find that our method consistently reduces retrieval-level errors such as noisy or incomplete evidence, which in turn leads to fewer summarization errors (e.g., omissions and query misalignment). We also clarify that LLM-based metrics (e.g., G-EVAL) are used only as supporting signals, not as primary evidence. This addresses Reviewer 2’s concerns (W1, W2, W5).
>
> We have also improved the human evaluation section by providing additional details on the annotation setup, sampling, and agreement measures (Appendix A.11), making the evaluation more transparent and reproducible.
>
> Finally, to address the concern about the link between segmentation quality and downstream performance, we include a new analysis (Appendix A.12) that explicitly examines the relationship between segmentation, retrieval quality, and summarization outcomes, including intermediate metrics such as topic purity and query coverage, along with correlation analysis. This directly responds to Reviewer 2’s concern (W3).
>
> It will be really appreciated if the meta-review is based on the revision and not on  the original submitted version. We believe this is in accordance with the core philosophy of TMLR.